# An Overview of Artificial Intelligence Application for Optimal Control of Municipal Solid Waste Incineration Process

Jian Tang [1,2,*] , Tianzheng Wang [1,2] , Heng Xia [1,2] and Canlin Cui [1,2]

1 Faculty of Information Technology, Beijing University of Technology, Beijing 100124, China; wangtz@emails.bjut.edu.cn (T.W.); xiaheng@emails.bjut.edu.cn (H.X.); cuicanlin@emails.bjut.edu.cn (C.C.)
2 Beijing Laboratory of Smart Environmental Protection, Beijing 100124, China
* Correspondence: freeflytang@bjut.edu.cn or freeflytang@126.com

**Abstract:** Artificial intelligence (AI) has found widespread application across diverse domains, including residential life and product manufacturing. Municipal solid waste incineration (MSWI) represents a significant avenue for realizing waste-to-energy (WTE) objectives, emphasizing resource reuse and sustainability. Theoretically, AI holds the potential to facilitate optimal control of the MSWI process in terms of achieving minimal pollution emissions and maximal energy efficiency. However, a noticeable shortage exists in the current research of the review literature concerning AI in the field of WTE, particularly MSWI, hindering a focused understanding of future development directions. Consequently, this study conducts an exhaustive survey of AI applications for optimal control, categorizing them into four fundamental aspects: modeling, control, optimization, and maintenance. Timeline diagrams depicting the evolution of AI technologies in the MSWI process are presented to offer an intuitive visual representation. Each category undergoes meticulous classification and description, elucidating the shortcomings and challenges inherent in current research. Furthermore, the study articulates the future development trajectory of AI applications within the four fundamental categories, underscoring the contribution it makes to the field of MSWI and WTE.

**Keywords:** municipal solid waste incineration; optimal control; artificial intelligence; modeling; control; optimization; maintenance

## 1. Introduction

Artificial intelligence (AI) [1] has become extensively integrated across various industries, including metallurgy, petrochemicals, and energy, emerging as the primary catalyst for intelligent manufacturing [2–4]. This transition from the third industrial revolution, characterized by automation, is advancing into the fourth industrial revolution, commonly referred to as Industry 4.0 [5,6]. This evolution is marked by the seamless integration of AI technologies. In response to the imperatives of industrialization and automation, industrial sites deploy a multitude of sensors to gather diverse process data [7]. Concurrently, advancements in the Internet of things (IoT), cloud computing, and big data analytics significantly augment the capacity and possibility of integrating AI into industrial processes [8,9].

Presently, the global annual growth rate of municipal solid waste (MSW) has surged from 8% to 10% [10]. MSW incineration (MSWI) technology stands as a pivotal waste-to-energy (WTE) method, offering an effective solution to challenges related to environmental sustainability [11]. As a typical industrial process [12,13], MSWI achieves WTE through a sequence of stages, encompassing fermentation, combustion, heat exchange, and gas cleaning [2,14]. In the fermentation stage, numerous uncertain biological reactions take place. The combustion stage is characterized by high-temperature chemical reactions involving solid, gas, and liquid phases, driven by heat flow forces. The heat exchange stage facilitates the conversion of heat energy into mechanical energy and subsequently into electric energy. The flue gas cleaning

stage employs physical and chemical principles to eliminate toxic and harmful substances from the flue gas. In addition to meeting its energy requirements, the MSWI process provides various forms of energy, including electricity and heat [15]. Furthermore, it ensures a reduced risk of environmental pollution emissions. Studies indicate that the MSWI process achieves remarkable rates, with mass reduction reaching 70%, volume reduction at 90%, and energy recovery reaching 19% [16,17]. Developing countries recognize the substantial economic and environmental protection potential of this process [18,19].

After half a century of development, the MSWI control system has undergone a transformative shift towards a large-scale, integrated, and intelligent direction. This evolution is attributed to the integration of automation technology, computer technology, and advancements in incineration equipment and processes [3]. Currently, operational, under-construction, and proposed MSWI plants predominantly employ grate furnace incinerators, high-parameter boiler power generation equipment, and progressive cumulative flue gas cleaning processes. The overarching objective is to facilitate the low-carbon transformation of enterprises, thereby enhancing economic efficiency and competitiveness [20,21]. Nevertheless, the composition and generation of MSW are influenced by various uncertainties and regional factors, encompassing societal, economic, and environmental aspects [22]. The utilization of large-scale operational equipment further complicates the achievement of efficient and stable control of the MSWI process [23,24]. Therefore, the development of intelligent optimal control for the MSWI process, with AI assistance, is currently in progress. Figure 1 illustrates the annual number of studies related to AI applications in the MSWI process within the Web of Science (WoS) database. The data span from 1996 to 2023, and the keywords employed include "modeling", "control", "optimization", and "maintenance".

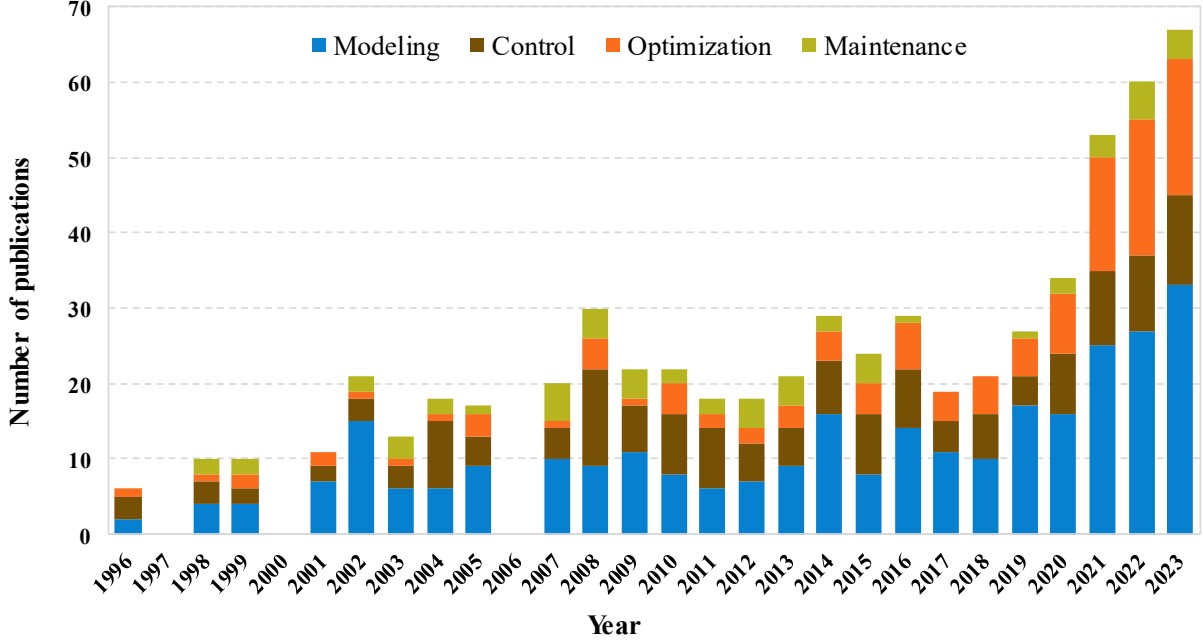

**Figure 1.** Number of annual publications of AI application for MSWI in the WoS.

Figure 1 illustrates a notable upswing in studies concentrating on the application of AI in MSWI in recent years. Notably, within this trend, modeling emerges as the predominant research direction, exhibiting a consistently increasing number of studies. The exploration of optimization has witnessed significant growth, particularly post-2021. In contrast, research on control and maintenance has maintained a relatively stable level, with studies on maintenance constituting a smaller proportion. Consequently, the investigation into the intelligent optimal control of the MSWI process, propelled by AI applications, is progressively evolving into a focal point of research.

The existing studies reveals a notable gap in the literature, as a comprehensive review of AI applications for intelligent optimal control of the MSWI process is currently absent. This study aims to systematically review the existing research in MSWI that incorporates AI techniques, intending to fill this gap. The contributions of this study are delineated as follows: (1) The review is undertaken from four key perspectives, namely modeling, control, optimization, and maintenance, with a specific focus on the intelligent control tasks of the MSWI process. The detailed summary of AI applications in these four aspects is crafted to provide valuable insights for researchers and practitioners. (2) A timeline map is incorporated to visually depict the evolution of various AI algorithms in the MSWI process, offering a clear representation of the AI application trends over time. (3) The review is intricately intertwined with the practical aspects of the MSWI process, and the applied AI algorithms are comprehensively discussed. Furthermore, the review identifies current challenges and proposes future directions, thereby contributing to the progression of AI applications in the field of MSWI.

The structure of this study is shown in Figure 2.

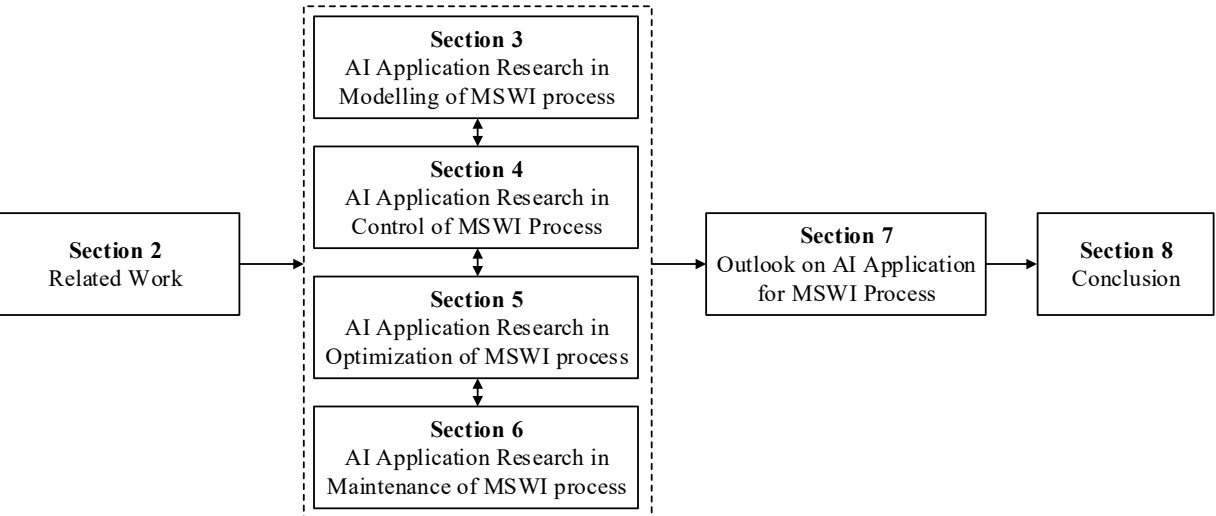

**Figure 2.** Structure of this study.

Section 2 introduces the literature review methodology, provides a detailed description of the MSWI process, and offers a brief overview of AI applications for optimal control. Subsequently, Sections 3–6 delve into the individual fields of AI application research, addressing modeling, control, optimization, and maintenance of the MSWI process. In Section 7, the focus shifts to an in-depth discussion of the prospects and outlook on AI applications within the MSWI process. Finally, Section 8 encapsulates the key findings and conclusions of this study.

## 2. Related Work

### 2.1. Methodology about the Literature Review

The literature reviewed in this study was systematically collected and processed from prominent scientific research databases, including WoS, Engineering Village, PubMed, and China National Knowledge Internet (CNKI). The retrieval time range spans from the establishment of each database to December 2023. Subsequently, the collected literature underwent a meticulous filtering process to exclude unrelated works. The refined literature was then categorized into four distinct sections based on AI applications in the MSWI process, namely modeling, control, optimization, and maintenance. Figure 3 illustrates the distribution of the literature across these four categories, providing an overview of the research landscape.

**92 literatures for reviewing AI application in MSWI optimal control**

**45 literatures for Modeling**
E.g., Meng, X.; Tang, J.; Qiao, J. NOx Emissions Prediction with a Brain-Inspired Modular Neural Network in Municipal Solid Waste Incineration Processes. *IEEE Transactions on Industrial Informatics* **2021**, 18 (7), 4622－4631.

**30 literatures for Control**
E.g., Leskens, M.; van Kessel, Lbm.; Bosgra, O. Model Predictive Control as a Tool for Improving the Process Operation of MSW Combustion Plants. Waste *Management* **2005**, 25 (8), 788－798.

**6 literatures for Optimization**
E.g., Cui, Y.; Meng, X.; Qiao, J. Multi-Condition Operational Optimization with Adaptive Knowledge Transfer for Municipal Solid Waste Incineration Process. *Expert Systems with Applications* **2024**, 238, 121783.

**11 literatures for Maintenance**
E.g., Tavares, G.; Zsigraiová, Z.; Semiao, V.; da Graca Carvalho, M. Monitoring, Fault Detection and Operation Prediction of Msw Incinerators Using Multivariate Statistical Methods. *Waste Management* **2011**, 31 (7), 1635－1644.

**Figure 3.** The number of literature studies in each category, e.g., [25–28].

Figure 3 reveals a notable concentration of research in the areas of modeling and control, while comparatively fewer studies focus on the optimization of the MSWI process using AI technology. It is crucial to highlight that optimization plays a pivotal role in achieving the sustainable development of the MSWI process. Despite the current emphasis on modeling and control, future research endeavors should recognize and address the significance of optimization for the overall efficacy and sustainability of MSWI operations.

*2.2. Description of MSWI Process in Terms of Optimal Control*

Figure 4 depicts the process flow of the grate-type MSWI process in Beijing.

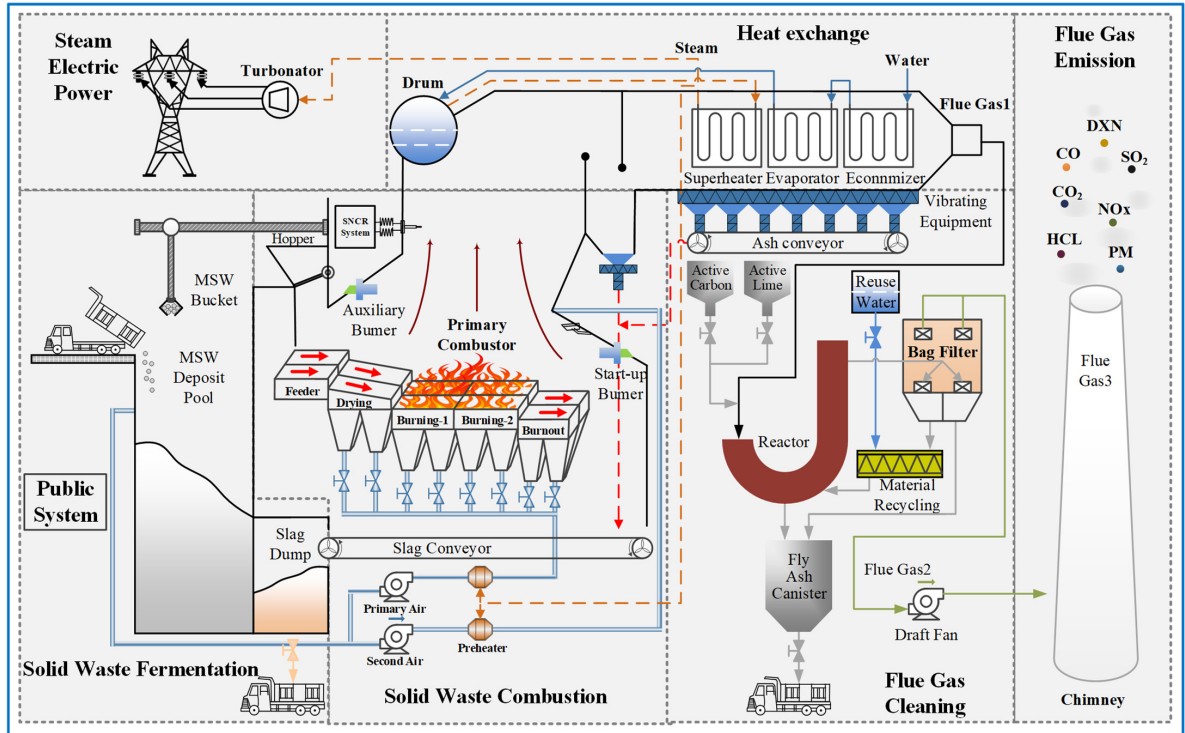

**Figure 4.** Process flow of an MSWI plant in Beijing. Note: Flue Gas1 denotes the flue gas at the furnace outlet. Flue Gas2 represents the flue gas at the inlet position of the induced draft fan. Flue Gas3 corresponds to the flue gas at the chimney outlet. Within the existing body of research, the predominant focus has been on Flue Gas1 and Flue Gas3.

Figure 4 provides an illustration of the MSWI process, which encompasses six distinct stages: solid waste fermentation, solid waste combustion, heat exchange, steam electric power, flue gas cleaning, and flue gas emission. The primary functions of each stage are outlined as follows:

(1) Solid-waste fermentation stage: Original MSW undergoes a 3–7 day biological fermentation in the MSW deposit pool to reduce water content that hinders combustion [29]. Following dehydration, the MSW achieves incineration readiness and is then transferred to the hopper before being pushed into the incinerator. This progression is facilitated by the feeder, marking the initiation of the solid waste combustion stage. The primary optimal control variable for this stage is the calorific value of the MSW.

(2) Solid waste combustion stage: During the solid waste combustion stage, the MSW transforms into high-temperature flue gas and solid residues through the coupled interaction of multiphases, including solid–gas–liquid, and multiple fields such as heat–flow–force. This stage is intricately divided into three substages: drying, burning, and burnout.

    (a) Drying substage: The total moisture content of MSW on the dry grate, comprising both surface and internal moisture, profoundly influences its ignition. Surface moisture gradually evaporates as the furnace temperature increases, reaching complete evaporation at 100 °C. Concurrently, internal moisture precipitates and absorbs mass heat energy with a further rise in furnace temperature. Consequently, the total moisture content of MSW closely correlates with the calorific value, exerting a notable impact on the combustion status and overall working conditions of the entire process.

    (b) Burning substage: From the ignition of MSW to intense luminescent heating, culminating in the conclusion of the oxidation reaction, the process involves robust oxidation, pyrolysis, and atomic group collision reactions. The strong oxidation reaction signifies the comprehensive reaction of the combustible components with oxygen. Concurrently, pyrolysis occurs under anaerobic or near anaerobic conditions, where thermal radiation energy disrupts or reorganizes the chemical bonds between the elements of carbon-containing polymer compounds. This leads to the precipitation of volatiles, subsequently oxidized. The atomic group collision reaction signifies the electronic energy transition of the atomic group, coupled with the rotation and vibration of the molecule, generating infrared thermal radiation, visible light, and ultraviolet light. This complex process ultimately shapes the flame. Hence, the reactions involved in the combustion process are intricate and variable, characterized by strong coupling between each other and the attributes of multireaction synchronous operation. Key manipulated variables for maintaining a stable combustion process include air volume and grate speed.

    (c) Burnout substage: Following combustion, the residual combustible components in MSW predominantly consist of coke. Subsequently, due to the high temperature and the presence of primary air, the oxidation reaction of coke with $O_2$ takes place, along with the gasification reaction of coke with $CO_2$, water vapor, and other substances. Inert substances, including gaseous CO, $H_2O$, and ash, gradually accumulate until all MSW on the grate transforms into ash. The combustion weakens until it is completely halted [30]. Consequently, this process is characterized by low flammability, heightened inert substances, a relatively high oxidant content, and a low reaction zone temperature. Extending the burnout substage typically proves effective in enhancing the thermal ignition reduction rate of MSW and improving the reduction level.

To ensure the decomposition and combustion of harmful substances in the flue gas, the "3T+E" principle is frequently employed [31]. This principle dictates that the furnace temperature should surpass 850 °C, the flue gas residence time must exceed 2 s, and the flue gas turbulence intensity and excess air coefficient should be maintained at appropriate

values. Key manipulated variables in this stage include the feed rate, grate speed, and air volume. The primary controlled variables encompass furnace temperature, flue gas oxygen content, steam flow, and combustion line.

(3) Heat exchange stage: The heat exchange stage unfolds in a series of sequential steps. Firstly, the high-temperature flue gas undergoes initial cooling through the water wall. Secondly, heat energy is effectively transferred to the boiler through a combination of radiation and convection, involving key components such as the superheater, evaporator, and economizer. Thirdly, within the boiler, the water undergoes a transformative process, turning into high-pressure superheated steam that enters the steam power generation stage. Finally, the flue gas temperature at the boiler outlet is fast reduced to 200 °C. Rigorous control of the cooling rate at this stage is essential. The primary manipulated variable is the boiler feed water volume, and the main controlled variable is the steam flow.

(4) Flue gas cleaning stage: The flue gas cleaning stage encompasses several crucial steps. Firstly, the selective noncatalytic reduction (SNCR) system initiates the removal of NOx at temperatures ranging from 850 °C to 1100 °C. Secondly, the semidry deacidification process effectively neutralizes acidic gases, including HCl, HF, $SO_2$, and heavy metals, through the injection of lime and water. Thirdly, activated carbon plays a pivotal role by adsorbing DXN and heavy metals present in the flue gas. Finally, the comprehensive purification process concludes as the particulate matter, neutralizing reactants, and adsorbates of activated carbon in the flue gas are systematically removed by the bag filter. The primary manipulated variables in this stage include the consumption of urea, activated carbon, lime, and other materials.

(5) Flue gas emission stage: In the flue gas emission stage, the discharged flue gas adheres to the national emission standards of diverse countries and is released into the atmosphere through the chimney, facilitated by the induced draft fan. Presently, environmental indicators of significant concern encompass pollutants such as particulate matter, NOx, $SO_2$, HCl, and CO.

### 2.3. AI in Modeling, Control, Optimization, and Maintenance of MSWI Process

Figure 5 illustrates the task function description of the MSWI process and its AI applications, covering modeling, control, optimization, and maintenance.

In this study, the application of AI concerning the task functions of the MSWI process is classified into modeling, control, optimization, and maintenance. The structure of the review is outlined as follows:

(1) Modeling: The AI application in the modeling of the MSWI process is subdivided into combustion process modeling and operational indices modeling. Combustion process modeling, elaborated in Section 3.1, focuses on data-driven modeling. Operational indices modeling is detailed in Section 3.2, covering environmental, product, and economic indices modeling.

(2) Control: The AI application in the control of the MSWI process is categorized into on-site control and off-site control. The review of existing research on on-site control is presented in Section 4.1, encompassing topics such as automatic combustion control, fuzzy rule control, and expert rule control. Research on off-site control is discussed in Section 4.2, covering PID parameter tuning and RBF neural network.

(3) Optimization: The AI application in the optimization of the MSWI process, focusing on manipulated and controlled variables, is predominantly discussed in Section 5. Particle swarm optimization (PSO) is highlighted as a significant algorithm in this field.

(4) Maintenance: The AI application in the maintenance of the MSWI process is categorized into three parts: recognition of flame status, qualitative detection of operational faults, and quantitative detection of operational faults. Recognition of flame status, utilizing random forest and deep forest classification, is introduced in Section 6.1. Qualitative detection of operational faults is discussed in Section 6.2, covering applications such as case-based reasoning, backpropagation neural network, and random

weight neural network. Quantitative detection of operational faults is presented in Section 6.3, including the application of principal component analysis (PCA) and partial least squares (PLS).

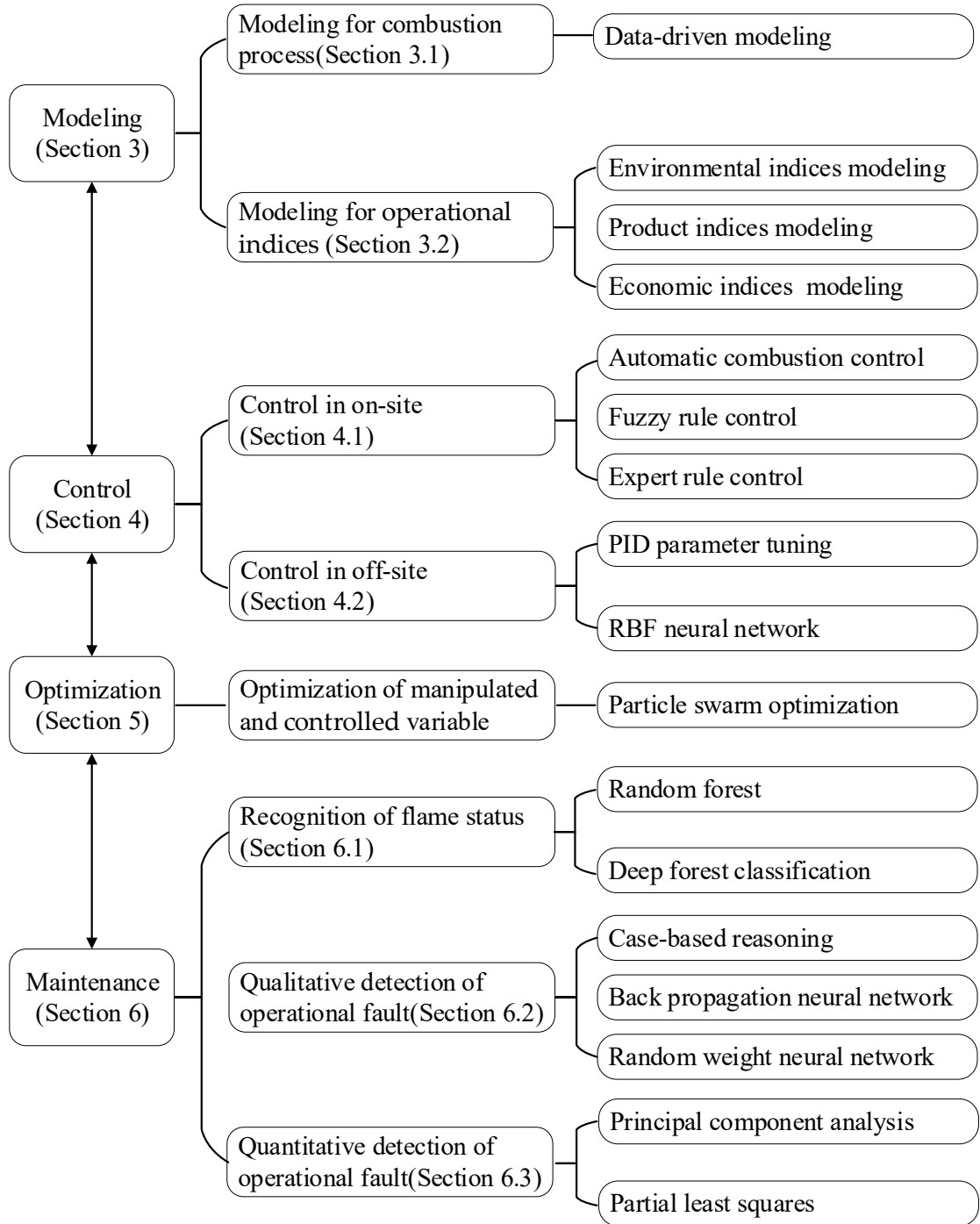

**Figure 5.** AI application in terms of the task function of the MSWI process.

*2.4. Development of AI Applications Research in the MSWI Process*

To visually illustrate the relationship among development, method, and application, Figure 6 provides a timeline summarizing the evolution of AI algorithms and their applications in the MSWI field. The timeline includes the proposed years of the methods, the first application in the MSWI field, and subsequent method applications. AI algorithms are cat-

egorized into four groups: machine learning, fuzzy logic, metaheuristic methods, and deep learning. It is important to note that only classic methods are presented in each category. Figure 6 shows as follows:

(1) Machine learning stands out as a prominent AI method in the application of the MSWI process. Figure 6 provides a comprehensive summary of machine-learning applications, encompassing neural network (NN), support vector machine (SVM), PCA, and tree-based model (TM). Within this domain, NN methods represent the most popular direction. Firstly, NN exhibits robust learning capabilities, allowing its application in various tasks such as control, modeling, and maintenance. Secondly, the flexible structure of NN permits adaptations based on specific operational requirements and conditions. Despite the earlier proposals of TM and SVM methods, their application in the MSWI process did not realize until 2017. Additionally, PCA is employed for feature extraction in modeling and monitoring, but its practical applications are relatively limited.

(2) Fuzzy logic (FL) is a well-established method renowned for controlling complex process systems. Consequently, FL has found application in the MSWI process since 1989. FL emerged as one of the most popular control methods between 2003 and 2005, extending its application to maintenance and modeling in the MSWI process. However, research on FL has gradually diminished in recent years, likely influenced by the emergence of NN and other methods. In response to this trend, researchers have introduced the fuzzy neural network (FNN) method by seamlessly combining FL and NN.

(3) PSO is a form of evolutionary algorithm categorized under metaheuristic methods. These methods demonstrate proficiency in searching for optimal parameters for models and controllers of the MSWI process. However, the application scope of metaheuristic methods is constrained by factors such as randomness and time cost.

(4) Deep learning (DL) was developed in 2006, rendering it relatively more novel compared to other methods. The applications of the DL method in the MSWI process were concentrated in 2021 and 2022. It is anticipated to undergo rapid development in future studies.

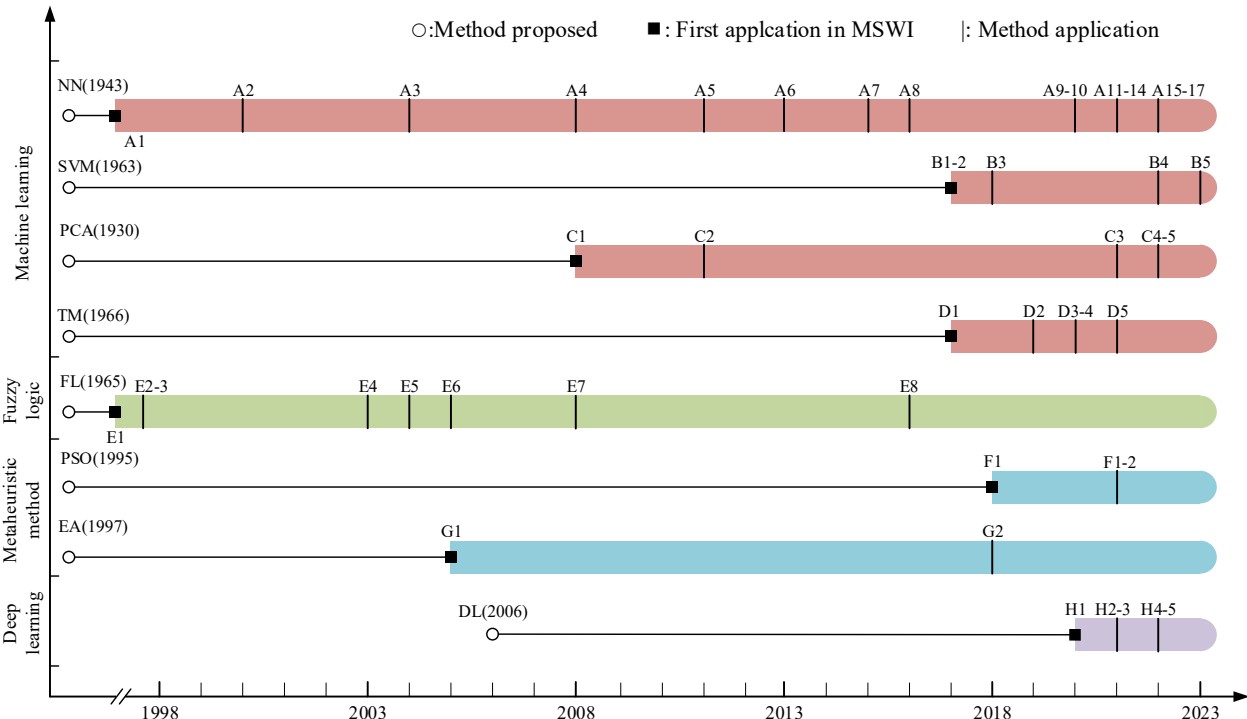

**Figure 6.** Timeline of AI algorithms and their applications in the MSWI process.

| Neural network (NN): | A15. (RBFNN)-Modeling-2022, [32] | Tree-based model (TM): | Particle swarm optimization (PSO): |
|---|---|---|---|
| A1. Control-1993, [33] | A16. (T-S FNN)-Modeling-2022, [34] | D1. (RF)-Modeling-2017, [35] | F1. Modeling-2021, [36] |
| A2. Modeling-2000, [37] | A17. (MNN)-Modeling-2022, [38] | D2. (RF)-Maintenance-2019, [39] | F2. Control-2018, [40] |
| A3. Modeling-2004, [41] | Support vector machine (SVM): | D3. (RF+GBDT)-Modeling-2020, [42] | F3. Optimization-2021, [43] |
| A4. Maintenance-2008, [44] | B1. Modeling-2017, [35] | D4. (RF)-Modeling-2020, [45] | Differential evolution (DE): |
| A5. (RBFNN)-Modeling-2011, [46] | B2. Modeling-2017, [47] | D5. (RF+GBDT)-Modeling-2021, [48] | G1. Optimization-2005, [49] |
| A6. Modeling-2013, [50] | B3. (LS-SVM)-Modeling-2018, [51] | Fuzzy logic (FL): | G2. Control-2006, [52] |
| A7. Maintenance-2015, [53] | B4. Modeling-2022, [54] | E1. Control-1989, [55] | Deeping learning (DL): |
| A8. Modeling-2016, [56] | B5. (LS-SVM)-Modeling-2023, [57] | E2. Control-1991, [58] | H1. (DBN)-Modeling-2020, [59] |
| A9. (FNN)-Modeling-2020, [60] | Principal component analysis (PCA): | E3. Maintenance-1994, [61] | H2. (Yolov5)-Modeling-2021, [62] |
| A10. (MNN)-Modeling-2020, [63] | C1. Maintenance-2008, [64] | E4. Control-2003, [65] | H4. (DFR-clfc)-Modeling-2021, [66] |
| A11. Modeling-2021, [67] | C2. Maintenance-2011, [28] | E5. Control-2004, [68] | H5. (IDFR)-Modeling-2022, [13] |
| A12. Modeling-2021, [69] | C3. Modeling-2021, [70] | E6. Control-2005, [71] | H7. (GAN)-Maintenance-2022, [72] |
| A13. (MNN)-Modeling-2021, [25] | C4. Modeling-2022, [32] | E7. Control-2008, [73] | |
| A14. (RWNN)-Maintenance-2021, [74] | C5. Modeling-2022, [54] | E8. (ANFIS)-Modeling-2016, [56] | |

## 3. AI Application Research in Modelling of MSWI Process

As a typical process industry, the MSWI process displays strong nonlinearity and coupling, involving numerous stages and variables. To precisely depict the AI application in the modeling of the MSWI process, this study categorizes it into two main areas: modeling for the combustion process and operational indices.

### 3.1. Modeling for Combustion Process

Typically, complex industrial processes use historical data to construct models for controlled objects, validating intelligent control algorithms [75–78]. This subsection is further divided into two parts, namely key controlled variables and auxiliary variables.

### 3.1.1. Key Controlled Variables

Key controlled variables in the combustion process encompass furnace temperature (FT), flue gas oxygen content (FGOC), steam flow (SF), and combustion line position (CLP), which refers to the position where the end of MSW becomes ash [79], et al.

(1)    Multi-input single-output (MISO) modeling

FT is typically measured using a thermocouple, serving as a vital parameter to characterize the stability of the combustion status and directly influencing pollutant emissions [50,80]. The establishment of a controlled FT model is a crucial prerequisite for achieving stable control and validating algorithms [46,81]. Existing studies on data-driven models include multimodel intelligent combination [82], T-S fuzzy neural network [60], and least squares-support vector regression (LS-SVR) [57]. However, these studies mostly focused on a single operating condition within a narrow range, highlighting the need for improvement in their adaptability.

FGOC refers to the coefficient of excess air, which can characterize the combustion status to a certain extent [83]. Measuring points for FGOC are typically installed at the outlet of the waste heat boiler (Flue Gas1) and the chimney (Flue Gas3). Sun et al. [70] proposed a weighted PCA and an improved long short-term memory network (LSTM) strategy for constructing a prediction model at the Flue Gas3 location, but further improvement is needed in modeling accuracy.

SF determines the recovery efficiency of the waste heat boiler and the power generation of the steam turbine [67]. Studies on predicting models for SF include: Gianto-Massi et al. [46] adopted a radial basis function (RBF) neural network based on adaptive Kalman filter parameter updating. Sun et al. [32] used RBF based on the average influence value algorithm for feature selection. Yang et al. [84] employed the LSTM algorithm, among others.

For CLP, Miyamoto et al. [85] proposed a quantitative method based on process data and flame images. However, research on constructing the mapping model between manipulated variables and CLP has not been reported.

Unfortunately, the aforementioned studies all employed data-driven methods to construct MISO soft sensing or prediction models, which are not control-oriented models for controlled objects. Consequently, it is challenging to support the research of optimal control algorithms.

(2)     Multi-input multi-output (MIMO) modeling

As a typical MIMO system, the coupling between manipulated and controlled variables in the combustion process is significant. Leskens et al. [86] constructed an ARX model for FGOC and SF. Furthermore, for FT, FGOC, and SF, Chen et al. [36] constructed a cascade transfer function model based on adaptive weight PSO; Ding et al. [34] built a T-S fuzzy neural network model; and Wang et al. [48] built a hybrid ensemble model of random forest (RF) and gradient boosting decision tree (GBDT), whose strategy diagram is shown in Figure 7.

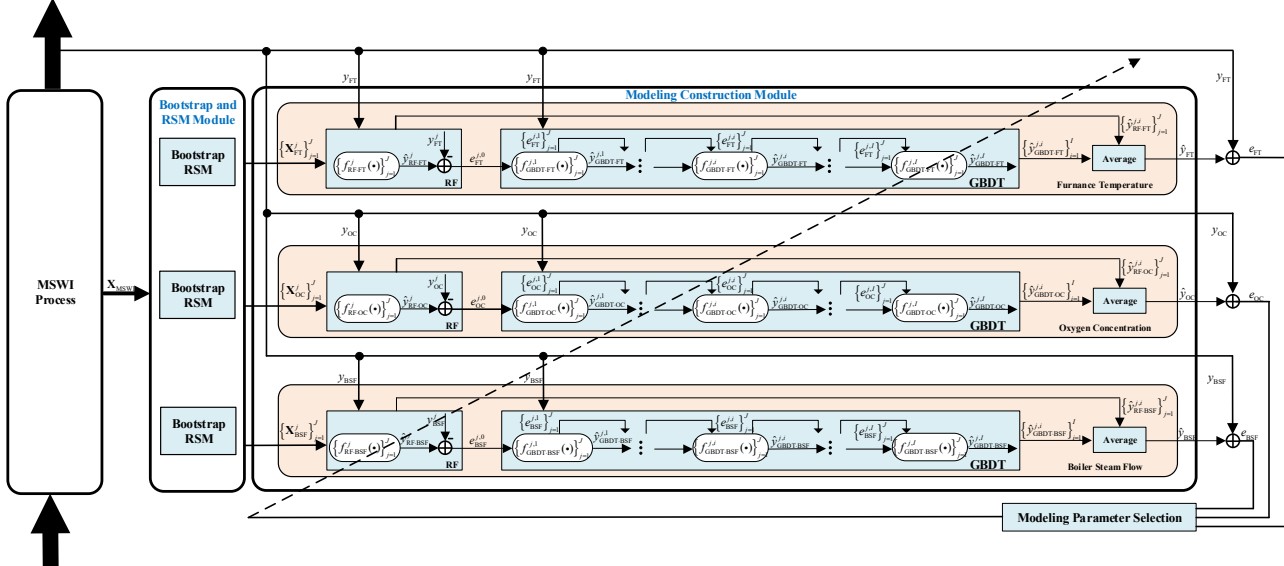

**Figure 7.** Strategy diagram of Ref. [48].

The studies mentioned above support research in optimal control; however, they encounter challenges such as poor modeling accuracy and an unresolved issue regarding the model's adaptability under various operating conditions.

In summary, the combustion process exhibits nonlinear and strong coupling characteristics. Building upon existing studies of other industrial processes [87,88], there is a need for in-depth research on the MIMO-controlled object model of the MSWI process, particularly focusing on its adaptability to complex operating conditions.

### 3.1.2. Auxiliary Variables

The stability of the combustion process is influenced by numerous auxiliary variables. However, this study specifically focuses on the calorific value of MSW and the thickness of the MSW layer.

(1)     Calorific value of MSW (CVMSW)

The CVMSW is a crucial factor in the combustion process. It directly influences the selection of manipulated strategies, including decisions on whether to add auxiliary fuel and in what quantity. Additionally, it impacts the operation, maintenance, management, and economic benefits of enterprises [89,90]. In addressing the challenge of direct detection difficulty, Chen [91] and Zeng et al. [92] employed the heat balance mechanism for estimation. Kessel et al. [93] initially constructed a soft sensing model based on process data. Subsequently, various data-driven models based on back propagation neural network (BPNN) [56,94–97], L-M backpropagation neural network [98], RBF [56], adaptive network-based fuzzy inference system (ANFIS) [56], FNN [69], and others have been successively introduced. Non-neural network soft sensing models encompass SVM [35], LS-SVM [51], and RF [35], among others. Additionally, You et al. [35] conducted a comparison of artificial neural network (ANN), ANFIS, SVM, and RF, with the results indicating that ANFIS

exhibited the best performance, followed by RF, while ANN performed less effectively. Recently, Xie et al. [62] introduced a real-time soft sensing model for calorific value based on deep learning and image recognition. Unfortunately, the truth samples used to construct a soft sensing model for the calorific value of MSW present challenges, including high acquisition costs, sparse samples, and a limited range of operating conditions. Combining the characteristics of modeling data is necessary to enhance generalization performance.

(2)   Thickness of the MSW layer (TMSWL)

The TMSWL undergoes dynamic changes throughout the combustion process and is closely related to the calorific value of MSW and SF. Therefore, it can also be considered as a controlled variable. Nuclear instruments are employed for direct detection, but they pose challenges such as high costs, complicated maintenance, and limited practicality. Given the aforementioned challenges in obtaining truth samples, acquiring soft sensing models mainly relies on indirect calculations using data such as air pressure, air volume, negative pressure, and grate area, considering the perspective of physical properties [99,100]. Hence, achieving more accurate and economical real-time detection methods needs further study.

The modeling research studies on the combustion process are summarized in Table 1.

**Table 1.** Summary of modeling research studies on the combustion process.

| Category | Object | Technology | | Benefit | Year | Literature |
|---|---|---|---|---|---|---|
| Key controlled variables modeling | FT | Multimodel intelligent combination | ♦ | Based on the decision tree C4.5, the algorithm can be selected for different datasets to build an ensemble model to improve the accuracy. | 2019 | [82] |
| | | T-S Fuzzy neural network | ♦ | The correlation between FT and input variables is obtained by using the internal weight of the neural network. | 2020 | [60] |
| | | Least squares-support vector regression | ♦ | Based on the principle of minimizing structural risk, the generalization ability and robustness are improved, and the over-fitting problem is avoided. | 2023 | [57] |
| | FGOC | Long short-term memory network | ♦ | The PSO algorithm is used to optimize the network hyperparameters to improve the model accuracy. | 2021 | [70] |
| | SF | Radial basis function networks | ♦ | The minimum resource allocation network technology is combined with the adaptive extended Kalman filter to update all the parameters of the network, so as to serve the MPC. | 2011 | [46] |
| | | Radial basis function networks | ♦ | The mean impact value algorithm is used to filter the features, which enhances the robustness and accuracy while reducing the model structure. | 2022 | [32] |
| | | Long short-term memory network | ♦ | The dynamic update of the model based on real-time data improves the prediction accuracy. | 2021 | [84] |
| | CLP | Neural network | ♦ ♦ | Based on the waste quality and quantity in different incinerator types and different seasons, an online learning method is proposed, which can select an optimized neural network. The control accuracy of pollutant emission is improved by using flame combustion image information. | 1996 | [85] |

**Table 1.** *Cont.*

| Category | Object | Technology | | Benefit | Year | Literature |
|---|---|---|---|---|---|---|
| | FGOC and SF | System identification | ♦ | Multiple datasets can be used to improve the model's accuracy to adapt to a variety of working conditions. | 2002 | [86] |
| | FT, FGOC, and SF | System identification | ♦ ♦ | A cascade structure is designed by simulating the actual industrial process. The optimization algorithm is used to identify the parameters and improve the accuracy of the model. | 2021 | [36] |
| | FT, FGOC, and SF | T-S Fuzzy neural network | ♦ | The complementary information between multiple tasks is used to accurately fit multiple controlled variables at the same time, which improves the dynamic adaptability of the model. | 2022 | [34] |
| | FT, FGOC, and SF | Decision tree algorithm | ♦ | The integration of RF and GBDT not only simplifies the model dimension but also improves the model accuracy. | 2021 | [48] |
| | | Estimation of waste heat balance | ♦ | The method is simple and easy to calculate. | 2017 | [91] |
| | | Estimation of waste heat balance | ♦ | The variables involved are easy to monitor and the estimated values can be calculated in real time. | 2019 | [92] |
| | | Mass balance | ♦ | It can be directly obtained by online measurement of the gas components $CO_2$, $H_2O$, $O_2$, and the $H_2O$ in the surrounding air. | 2002 | [93] |
| | | Back propagation neural network, Radical basis function neural network, and Adaptive neural fuzzy inference system | ♦ | The method is simple, easy to implement, and low-cost. | 2016 | [56] |
| | | Back propagation neural network | ♦ | Based on the correlation analysis, the partial correlation coefficient is obtained, and then the model is established. Compared with the multiple linear regression model, the accuracy is improved. | 2002 | [94] |
| Auxiliary variables modeling | CVMSW | Back propagation neural network | ♦ | The accuracy of the model is improved by determining the network hyperparameters upon experiments and analysis. | 2003 | [95] |
| | | Back propagation neural network | ♦ | For inaccurate, contradictory, and erroneous data, the neural network model has stronger fault tolerance than the physical component model, and then obtains higher accuracy results. | 2010 | [96] |
| | | Back propagation neural network | ♦ | The genetic algorithm is used to optimize the network parameters, thereby improving the accuracy. | 2012 | [97] |
| | | L-M backpropagation neural network | ♦ | Based on the element content, the accurate prediction of the high calorific value of waste was realized. | 2010 | [98] |
| | | Fuzzy neural network | ♦ | Based on MI and PSO, the input features of the model are screened to reduce the computational complexity of the model. | 2021 | [69] |
| | | Back propagation neural network, support vector machine, adaptive neuro-fuzzy inference system, and random forest | ♦ ♦ | Based on expert experience, the calorific value of waste is classified. The PSO algorithm is used to optimize the model parameters, and then a model with high accuracy is established. | 2017 | [35] |

**Table 1.** *Cont.*

| Category | Object | Technology | Benefit | Year | Literature |
|---|---|---|---|---|---|
| | | Least-square support vector machine | ◆ The genetic algorithm is used to optimize the model parameters. <br> ◆ The sensitivity analysis experiment was carried out on the input characteristics. The results show that the percentage of carbon has the deepest influence on HHV prediction. | 2018 | [51] |
| | | Deep learning | ◆ It is proposed to establish a waste image database to support the combination of image recognition technology and deep learning to achieve calorific value prediction. | 2021 | [62] |
| | TMSWL | Soft sensing model | ◆ The parameters such as air pressure, negative pressure, grate area, air volume and temperature are used to estimate. | 2022 | [99] |
| | | Soft sensing model | ◆ The thickness of the MSW layer is estimated based on the MSW composition and grate movement. | 2022 | [100] |

### *3.2. Modeling for Operational Indices*

### 3.2.1. Environmental Indices Modeling

The indices related to environmental protection encompass numerous variables, among which particulate matter and emission concentrations of NOx, $SO_2$, HCl, HF, and $CO_2$ can be detected online through the continuous emission monitoring system (CEMS). The emission concentration of toxic heavy metals and organic pollutants, such as dioxin (DXN) and volatile organic compounds (VOCs), is primarily determined through offline testing conducted in the laboratory [101]. The subsection is divided into two categories: the prediction model for easily detectable indices and the soft sensing model for difficult-to-detect indices.

(1) Prediction model for easily detectable indices

Considering the reliability of the CEMS system and the requirement for intelligent optimal control, it is essential to construct a prediction model for easily detectable indices.

For NOx, Matsumura et al. [102] initially employed system identification to construct a NOx emission model and utilized its output as a feedback signal to control the amount of NH3 injected. Additionally, Huselstein et al. [103] used continuous-time system identification [104] to establish a multitransfer model of NOx emissions with FGOC and secondary air volume as inputs. They analyzed the effects of manipulated variables, such as air volume and feed volume, on NOx emissions. Subsequently, many researchers utilized machine-learning algorithms to build NOx emission prediction models, including BPNN [41], RBF [25,63], and LSTM [38]. Nevertheless, practical verification of the aforementioned models on actual MSWI plants remains to be conducted.

As one of the toxic gases produced by the MSWI process, carbon monoxide (CO) must be strictly controlled [105]. Additionally, CO is directly related to DXN [59]. The standard of a half-hour average is generally adopted due to the noticeable spike phenomenon in CO emission concentration [106]. Zhang et al. [107] proposed a CO emission prediction method based on reduced-depth features and a long short-term memory (LSTM) optimization strategy. This method comprises three parts: a particle design for the reduced-depth feature and LSTM optimization, a fitness function design for the reduced-depth feature and LSTM optimization, and optimization based on PSO. The strategy diagram is shown in Figure 8.

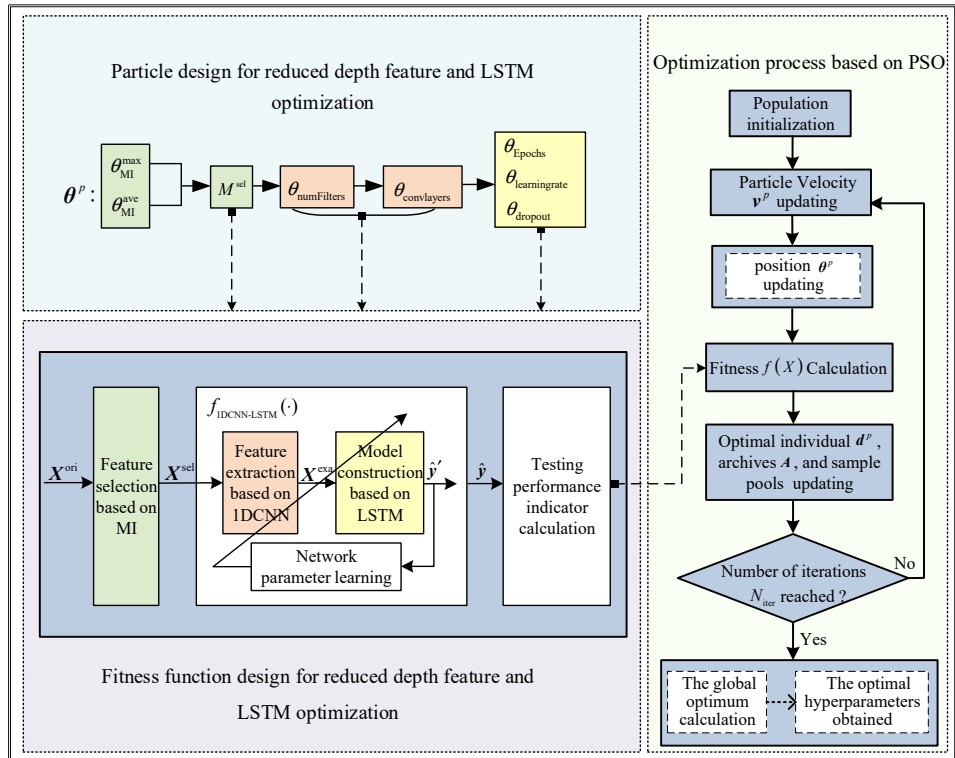

**Figure 8.** Strategy diagram of Ref. [107].

The results demonstrate that the machine learning-based prediction models mentioned above can effectively predict environmental indices in specific scenarios. Unfortunately, prediction models for particulate matter and acidic gases such as HCl and HF have not been reported yet [108]. Most of the existing studies have utilized software, such as computational fluid dynamics, for numerical simulation [109,110], and subsequently provided support for optimizing process design and analyzing mechanisms. Notably, studies on carbon emissions from the MSWI process have not been reported.

(2)  Soft sensing model for difficulty-to-detect indices

In consideration of environmental indices that cannot be detected online, this study focuses solely on reviewing DXN, which contributes to the "NIMBY effect" of MSWI plants [111]. From the perspective of the generation mechanism, DXN reactions, including formation, decomposition, resynthesis, and adsorption, are distributed throughout the entire process. These related physical and chemical reactions occur within a short timeframe. There is a "memory effect" that has not been adequately explained [112]. Obtaining complete modeling samples is challenging due to the time, labor, and economic costs associated with on-site sampling and laboratory testing.

DXN concentration is primarily determined through laboratory tests. After collecting samples at the site to be detected over an extended period, inspectors transport the samples to the laboratory for testing, ultimately obtaining the DXN concentration at the sampling time. The detection process is illustrated in Figure 9.

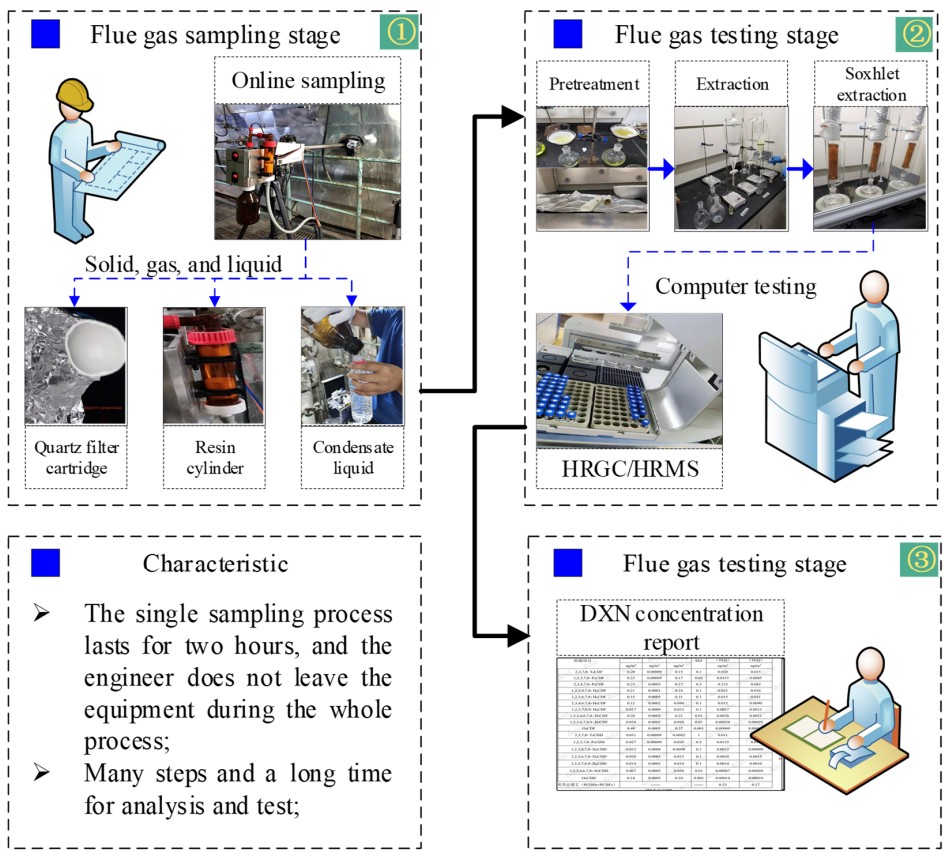

**Figure 9.** The detection process of DXN concentration [113].

Clearly, the aforementioned detection process of DXN concentration lacks real-time capabilities, making it challenging to support the intelligent optimal control of the MSWI process with dynamic changes in operating conditions. Therefore, establishing a soft sensing model for DXN becomes a popular approach to guide the research on optimal control.

Hasberg et al. [114] established the mapping relationship between flue gas temperature, CO, and DXN concentration. Chang et al. [115] developed a multiple linear regression analysis model, indicating a linear mapping relationship between DXN concentration, combustion chamber temperature, and CO concentration when FGOC was 7%. Building upon this, a linear mapping model between DXN concentration and flue gas flow, furnace temperature, and manipulated variables was established. Additionally, Ishikawa et al. [116] constructed a linear model for DXN concentration with FGOC, proportion of primary air volume, and total air volume as inputs through regression analysis of actual data. The aforementioned models face challenges in accurately describing the nonlinear relationship between their inputs and outputs.

Studies utilizing neural network algorithms include BPNN based on genetic programming for parameter identification [117], and BPNN based on the genetic algorithm to optimize parameters [37]. The studies utilizing the SVR algorithm include SVR based on the mechanism for feature selection [47] and selective ensemble SVR based on PCA in terms of subregion extraction and selection of latent features [54]. The studies based on the decision tree algorithm include a hybrid ensemble based on RF and GBDT [42] and RF based on sample transfer learning [45]. The feature learning ability of the aforementioned classical machine-learning model needs further enhancement.

Currently, existing studies primarily concentrate on the DXN concentration model of the G3 flue gas position, and they exhibit the following common problems: (1) Limited performance improvement of the model due to sparse samples; (2) Insufficient studies that integrate MSWI process and DXN mechanism characteristics; (3) Difficulty in fully

supporting DXN emission reduction control based on the generation mechanism with the existing models.

In summary, it is essential to conduct in-depth exploration and research on the environmental indices for the MSWI process.

### 3.2.2. Product Indices Modeling

The product indices of the MSWI process are significantly different from those of industrial processes such as mineral processing and petrochemical. The reason is that the MSWI process lacks commodity attributes [118]. In this study, the product indices of the MSWI process are determined as fly ash yield, heat reduction rate, and combustion efficiency.

(1)    Fly ash yield

The production of fly ash in the MSWI process results from the combustion of MSW, the generation of particles during desulfurization and deacidification, and the presence of activated carbon after adsorption of DXN and heavy metals, among other contributors [119,120]. This fly ash poses a potential threat to the sustainable development of both human societies and the ecological environment [121]. Given the limitations of air pollution control devices (APCDs) technology and the increasingly stringent environmental protection emission policies, coupled with the challenges in effectively controlling the yield of fly ash, the primary focus of research in both industry and academia is predominantly on the harmless treatment [122,123] and resource utilization [124,125], among other considerations [126]. Consequently, addressing the complex issues related to modeling, prediction, and intelligent optimal control remains a significant challenge.

(2)    Heat reduction rate (HRR)

The HRR is defined as the percentage reduction in slag quality after burning compared to its original state. This index plays a crucial role in assessing the incineration effectiveness and the reduction rate of MSW capacity [127]. According to relevant national standards, the upper limit for the HRR is set at 5%. Currently, measurement involves an offline testing mode, encompassing on-site sampling, transportation, and sample delivery, as well as weighing, drying, burning, cooling, and subsequent laboratory analysis [128]. To address the challenges of offline testing, Luo et al. [129] developed online detection equipment. However, the harsh working environment posed difficulties in maintaining stable operations over an extended period. Another approach by Sun et al. [130] involved associating the appearance characteristics of slag with its heat reduction rate, although a corresponding mapping model was not constructed. While these studies present initial exploratory ideas for reliable online detection, further research is needed to establish a robust and comprehensive methodology.

As of now, practical industrial applications heavily depend on expert experience to regulate the HRR. Common strategies involve extending the combustion time on the grate and implementing oxygen-enriched combustion [131,132].

(3)    Combustion efficiency

Combustion efficiency is defined as the ratio of the heat produced during fuel combustion to the low calorific value released by complete fuel combustion under adiabatic conditions. Unfortunately, there is no existing research on this aspect. The Chinese standard for pollutant control in hazardous waste incineration [(GB 18484-2020)] [133] defines it as the percentage of $CO_2$ concentration in the flue exhaust gas to the sum of $CO_2$ and CO concentration. Previous studies on coal combustion and cocombustion of other fuels [134,135] have demonstrated that combustion efficiency serves as a quantitative measure of combustion status. Generally, higher combustion efficiency is deemed favorable. However, it may conflict with CO concentration and carbon reduction, underscoring the necessity for multi-objective optimization research.

In summary, there are currently no relevant reports on product indices, hindering optimal control research in the field of the MSWI process. Theoretically, optimizing control over product indices has the potential to reduce the operating costs of MSWI plants.

### 3.2.3. Economic Indices Modeling

The economic viability of an MSWI plant is predominantly driven by MSW processing fees and on-site power generation. Given the inherent environmental characteristics of the MSWI process, its rated capacity and turbine power generation exhibit relative inflexibility. Consequently, maintaining these parameters within optimal limits is crucial to ensure maximum returns. Typically, the power generation of an MSWI plant ranges from 0.3 to 0.7 MWh/t [136]. Under normal operating conditions, strategies aimed at maximizing power generation include: (1) Reducing processing capacity when the calorific value of MSW is high; (2) Increasing processing capacity when the calorific value is moderate; (3) Significantly enhancing MSW processing capacity when the calorific value is low. However, due to process constraints, power generation efficiency must decrease as MSW processing capacity increases. Currently, both heat exchange efficiency and combustion efficiency witness an increase. To optimize energy utilization, additional heat energy is directed towards heating primary and secondary air, as well as other stages requiring heat energy. Consequently, the mentioned economic indices necessitate a redefinition in research focusing on optimal control for the MSWI process. Presently, there are no documented studies on modeling and prediction in economic indices.

The modeling research studies on operational indices are summarized in Table 2.

**Table 2.** Summary of modeling research studies on operational indices.

| Category | Object | Technology | | Benefit | Year | Literature |
|---|---|---|---|---|---|---|
| Environmental indices | NOx | System identification | ♦ | It can not only compensate the delay time of the detection device, but also the whole process. | 1998 | [102] |
| | | System identification | ♦ | A continuous-time MISO reduced-order model is constructed. | 2002 | [103] |
| | | Back propagation neural network | ♦ | The number of hidden layer nodes is determined by dynamic construction method. | 2004 | [41] |
| | | Radial basis function neural network | ♦ | The complex task is decomposed into submodels to obtain a more accurate prediction model. | 2020 | [63] |
| | | Radial basis function neural network | ♦ | The self-organizing and competitive integration strategies are used to construct the submodel to enhance the generalization performance and efficiency. | 2021 | [25] |
| | | Long short-term memory | ♦ | A cooperative decision strategy is designed to ensure the generalization performance of modular model. | 2023 | [38] |
| | CO | Long short-term memory | ♦ | The PSO algorithm is used to adaptively reduce depth features and hyperparameters. | 2024 | [107] |
| | DXN | Numerical modeling | ♦ | The flow-and temperature distribution and the residence-time behavior are obtained. | 1989 | [114] |
| | | Linear regression | ♦ | Dummy variables are included to further provide the selective capability of different process. | 1995 | [115] |
| | | Linear regression | ♦ | Based on the static analysis of the model, suggestions for minimizing DXN concentration are given. | 1997 | [116] |
| | | Back propagation neural network | ♦ | The genetic programming model is used to screen out nonlinear models as well as identify the system parameters simultaneously in a highly complex system based on a small set of samples. | 2000 | [117] |

**Table 2.** *Cont.*

| Category | Object | Technology | | Benefit | Year | Literature |
|---|---|---|---|---|---|---|
| | | Back propagation neural network | ♦ | The genetic algorithm is used to optimize the parameters to improve the accuracy of the model. | 2008 | [37] |
| | | Support vector regression | ♦ | The inputs of the model are determined based on the mechanism and the correlation analysis of working conditions and conventional pollutants. | 2017 | [47] |
| | | Least squares-support vector machine | ♦ | The optimal selection algorithm based on branch and bound, and the information entropy weighting algorithm based on prediction error are used to adaptively select and weigh the candidate submodels. | 2022 | [54] |
| | | Random forest and gradient boosting decision tree | ♦ | The RF is used to reduce the model dimension, and then the GBDT algorithm is used to improve the model accuracy. | 2020 | [42] |
| | | Random forest | ♦ | The prediction errors are used to cyclically calculate the weight of the source and target domain samples. | 2020 | [45] |
| Product index | HRR | Equipment | ♦ | The automatic measurement is realized with less manual intervention, and the analysis efficiency is improved. | 2021 | [129] |
| | | Image recognition | ♦ | Based on the slag image, a reference card of slag color gradient marked with heat reduction rate is generated to guide related operations. | 2022 | [130] |

## 4. AI Application Research in Control of MSWI Process

Research indicates that the linchpin for ensuring the stable operation of the entire MSWI process resides in the incinerator [137]. The effective control of the combustion process, characterized by multiple variables, strong coupling, and nonlinearity, has perennially posed a central challenge in both industrial application and academic research. The ensuing review is structured around the dual perspectives of on-site and off-site control, to demarcate the boundary more clearly between industrial application and academic research and foster collaboration to address existing gaps [138].

### 4.1. Control in On-Site

In general, the automatic combustion control (ACC) system can achieve the automatic control of the combustion process under stable calorific value conditions of MSW and normal working circumstances [33]. However, significant manual intervention is required during abnormal conditions, including fluctuations in composition and calorific value due to insufficient mixed fermentation of MSW, steam flow falling below the rated value resulting in furnace temperature decreases, steam flow surpassing the rated value causing furnace temperature increases, and maintenance cycles of incineration equipment. Given these challenges, the industry has undertaken research to enhance the system.

#### 4.1.1. Research of ACC System

Schuler et al. [139] employed an infrared thermal imager to detect furnace temperature and its fluctuation information, enhancing rapid response in the fine-tuning process. Miyamoto et al. [140] used a neural network to construct a combustion status recognition model, utilizing its output as feedback information for the ACC system, resulting in an effective reduction in CO concentration. Zipser et al. [141] detected temperature information of MSW, flue gas, and flame through infrared image analysis to aid combustion control. To address fluctuations in furnace negative pressure caused by grate turnover, Zeng et al. [142] implemented a control scheme based on a filtering algorithm to ensure the stability of furnace temperature. For optimal combustion, Xu et al. [143] designed a closed-loop control strategy for steam flow to adapt to changes in MSW calorific value, achieving prolonged stable operation. Wang et al. [144] introduced denitrification, lime

slurry, emission factors, and other elements to the ACC system, achieving preliminary localization improvement.

4.1.2. Research of Non-ACC System

Concerning furnace temperature, Ono et al. [55] applied fuzzy rule control to an MSWI plant in Japan. Shen et al. [65] summarized expert experience as fuzzy control rules and implemented them in an MSWI plant in Shenzhen. Carrasco et al. [145] developed a combustion control system based on expert rules for an MSWI plant in Spain. However, rule-based control systems face challenges in maintaining stable operation in the presence of frequent fluctuations in operating conditions.

Despite the extended use of the introduced ACC system in developing countries over many years, MSWI plants still operate at a fundamental control level. Particularly in instances of damage to detection instruments and equipment, there is a greater reliance on manual control modes. Clearly, this impedes the achievement of long-term stable and optimal operation.

The research studies on control in on-site are summarized in Table 3.

**Table 3.** Summary of research studies on control in on-site.

| Category | Object | Technology | | Benefit | Year | Literature |
|---|---|---|---|---|---|---|
| ACC system | FT | Thermography-assisted combustion control system | ♦ | It can quickly obtain the temperature distribution in the furnace to reduce the response time, thereby reducing the fluctuation of parameters. | 1994 | [139] |
| | Whole process | Fuzzy system and Neural network | ♦ | Based on process data and flame images, a combustion state recognition model is established to assist ACC system decision making. | 1998 | [140] |
| | Whole process | Infrared image analysis instrument | ♦ | On-line acquisition and analysis of combustion images are realized. | 2006 | [141] |
| | Negative pressure | Expert experience | ♦ | While improving the negative pressure monitoring of the furnace to suppress the fluctuation of the negative pressure control system, the ACC scheme of the leachate recirculation flow control system is designed. | 2004 | [142] |
| | Whole process | Expert experience | ♦ | The controller performance requirements are low. | 2017 | [143] |
| | Pollutant | Expert experience | ♦ | The pollution emission data are added to the ACC system to intervene in advance to reduce emissions. | 2019 | [144] |

**Table 3.** *Cont.*

| Category | Object | Technology | | Benefit | Year | Literature |
|---|---|---|---|---|---|---|
| Non-ACC system | Whole process | Fuzzy logic | ♦ | The fuzzy logic rules of monitoring and control are developed through expert experience. | 1989 | [55] |
| | FT | Fuzzy logic | ♦ | The adaptability of the incinerator to the calorific value of MSW is improved. | 2003 | [65] |
| | FT | Fuzzy logic | ♦ | Knowledge is modularized in the form of rules and events to deal with different situations. | 2006 | [145] |

*4.2. Control in Off-Site*

The academic community has conducted numerous studies on key controlled variables from both single-input and single-output (SISO) and multi-input and multi-output (MIMO) perspectives.

4.2.1. SISO Control

(1)    Furnace temperature (FT)

Given the challenges faced by the introduced ACC system in China, researchers have undertaken extensive studies to address these issues. In the domain of fuzzy control, Qian et al. [33] compensated feeder control based on fuzzy rules using the MSW water content estimation model. Shen et al. [71] introduced a fuzzy rule controller with a self-tuning factor, demonstrating its capability to control furnace temperature stably. Based on Ref. [65], Chang et al. [146] designed a fuzzy rule controller with an adaptive weighted factor, highlighting its effective control. Considering practical issues such as real-time requirements and computational memory consumption, Wang et al. [68] proposed a hierarchical fuzzy rule control strategy with an optimized quantization factor and self-tuning scaling factor. A notable feature is that the correction factor can be selected based on the operating condition. Employing traditional PID control, Dai et al. [73] introduced a fuzzy adaptive PID controller to enhance the system's anti-interference ability, flexibility, and adaptability. He et al. [147] proposed an RBF-based PID parameter dynamic adjustment strategy to suppress disturbances. Additionally, Ni [148], Xiao [149], and Wu [150] et al. introduced a human-simulated intelligent controller (HSIC), simulating the cognitive mechanism and operational behavior of domain experts. Building on this, Wu et al. [40] proposed a PSO-based HSIC temperature controller.

The studies above have yielded satisfactory results, but the quantity of controlled variables in these studies is typically singular. This poses challenges in addressing the strong coupling characteristics inherent in the MSWI process.

(2)    Flue gas oxygen content (FGOC)

Sun et al. [151] introduced an RBF-based model predictive controller. After conducting a stability analysis of the control system, its effectiveness was verified through simulation.

(3)    Steam flow (SF)

Chen et al. [152] and Yang et al. [153] employed a fuzzy rule controller with grate speed as the manipulated variable, demonstrating a significant reduction in fluctuations caused by abnormal operating conditions. Watanabe et al. [154] adopted a feedback control strategy with a fixed time window to achieve stability control. Furthermore, Falconi et al. [155]

introduced a stable closed-loop control system based on a linear quadratic regulator, verifying its effectiveness through simulation experiments.

### 4.2.2. MIMO Control

(1)    Double input and double output

For the simultaneous control of steam flow and flue gas oxygen content, Leskens et al. [26] introduced a linear model predictive control (LMPC) strategy, demonstrating that the errors in both the manipulated and controlled variables were superior to those in traditional combustion control systems. However, the LMPC strategy encounters challenges when confronted with strong nonlinear problems. In response, Leskens et al. [156] presented a nonlinear model predictive control (NMPC) strategy aimed at estimating optimal air and material distribution across the rolling time domain. Additionally, they [157] introduced a PID control strategy that integrates components of the two loops, showcasing effective enhancement in tracking characteristics and a noteworthy improvement in the economic benefits of MSWI plants. In a similar vein, Ding et al. [158] proposed a self-organizing fuzzy neural network controller based on multitask learning for simultaneous control of furnace temperature and flue gas oxygen content. Nevertheless, its applicability is constrained to a single operational condition.

(2)    Triple input and triple output

For the concurrent regulation of furnace temperature, steam flow, and flue gas oxygen content, Ding et al. [159] introduced a multiloop PID controller utilizing a quasidiagonal recurrent neural network. This controller exhibits the capability to dynamically adjust its parameters in response to error signals. Moreover, Wang et al. [160] presented a multiple input multiple output control method founded on a single neuron adaptive PID. The accuracy and efficacy of this proposed method were validated through the analysis of real industrial data. Figure 10 illustrates the schematic representation of the control strategy.

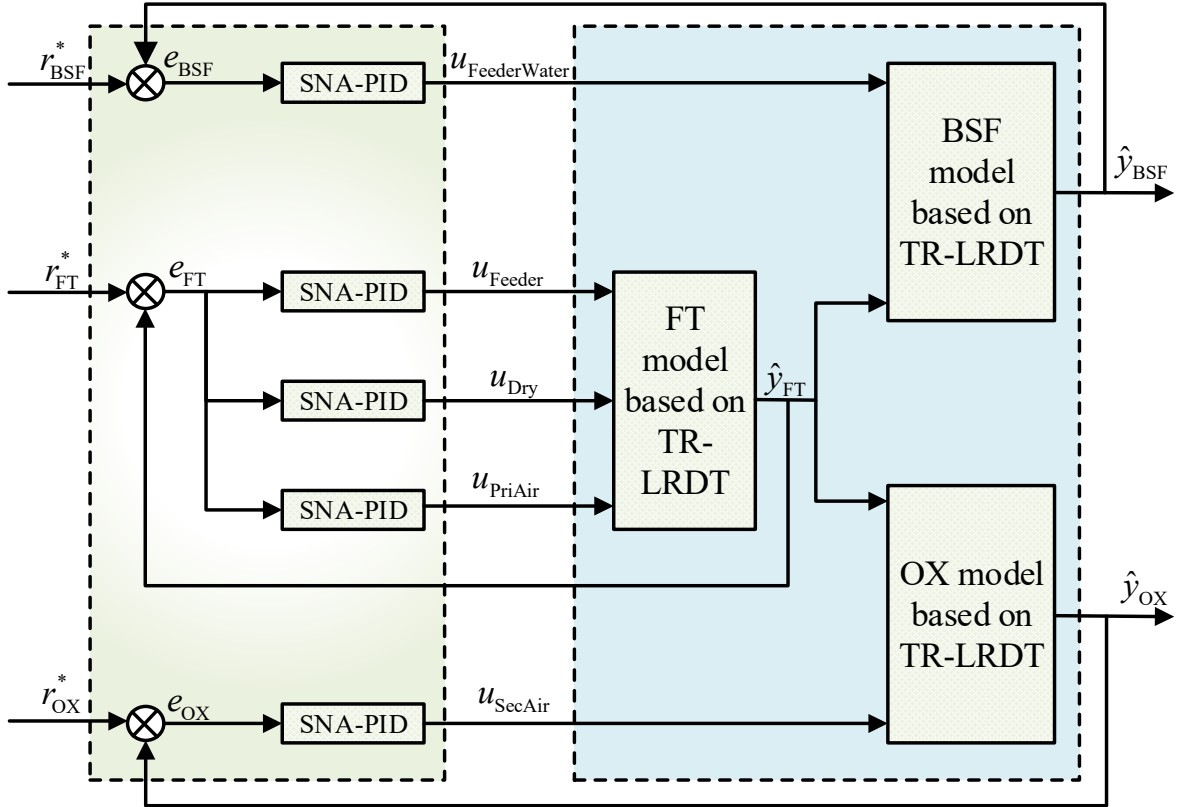

**Figure 10.** The diagram of control strategy in Ref. [160].

Nevertheless, the applicability of the aforementioned studies is constrained to specific operational conditions, highlighting the imperative to augment their universality.

These studies underscore the divergence in research focus between industrial applications and academic research. Clearly, employing AI for on-site or closed-loop control of the MSWI process not only addresses the issue of operating condition fluctuations arising from manual control modes and the expertise of domain experts but also contributes to cost reduction, enhanced energy efficiency, and reduced pollution emissions. Consequently, AI-assisted control is poised to become the prevailing trend in future research. However, the current AI algorithms researched face challenges in direct application due to the closure of the DCS system and safety requirements within MSWI enterprises. Therefore, the establishment of a hardware-in-loop simulation experimental platform is deemed necessary for testing and validating AI algorithms in industrial settings [138]. Clearly, the challenge lies in conducting more in-depth academic research in terms of application to enhance its universality and applicability to the actual industry, posing a complex problem that requires further exploration.

Table 4 provides a summary of research findings concerning control in off-site.

**Table 4.** Summary of research studies about control in off-site.

| Category | Object | Technology | | Benefit | Year | Literature |
|---|---|---|---|---|---|---|
| SISO | FT | Fuzzy logic | ◆ | The real-time estimation of wastewater content is realized to compensate the controller action error. | 1993 | [33] |
| | FT | Fuzzy logic | ◆ | The control rule factor optimized by the correction algorithm improves the ability of the control system to interference responds. | 2005 | [71] |
| | FT | Fuzzy logic | ◆ | Based on summarizing the control rules according to the weight variation law of temperature deviation and temperature deviation change rate, the adaptive weighting factor of input variables is introduced to improve the adaptive ability of the control system. | 2004 | [146] |
| | FT | Fuzzy logic | ◆ | To deal with the problem of rule explosion, a hierarchical fuzzy controller is proposed, and the online learning and correction of control parameters and control rules are realized. | 2004 | [68] |
| | FT | Fuzzy logic | ◆ | The fuzzy logic is used to automatically adjust the PID parameters, thereby improving the adaptability. | 2008 | [73] |
| | FT | Radial basic function, and Event-trigger | ◆ ◆ | The online adjustment of PID controller parameters is realized based on RBF. The event triggering method is used to reduce the update frequency of the controller. | 2022 | [147] |
| | FT | Human-simulated intelligent controller | ◆ | The control strategy is closer to the actual needs of the plant. | 2013 | [148] |
| | FT | Human-simulated intelligent controller | ◆ | The overshoot can be effectively suppressed for uncertain disturbances. | 2015 | [149] |
| | FT | Human-simulated intelligent controller | ◆ | No need for accurate theoretical model support. | 2016 | [150] |
| | FT | Human-simulated intelligent controller | ◆ | PSO is used to tune the controller parameters. | 2018 | [40] |
| | FGOC | Radial basis function, Model predictive control | ◆ ◆ | The adaptive fuzzy C-means is used to determine the network parameters. The prediction model parameters are adjusted online by an adaptive update strategy. | 2023 | [151] |

**Table 4.** *Cont.*

| Category | Object | Technology | Benefit | Year | Literature |
|---|---|---|---|---|---|
| | SF | Fuzzy logic | ♦ It has the potential of about a 10% increase in the capacity of MSW processing and electricity generation. | 1995 | [152] |
| | SF | Fuzzy logic | ♦ The controller can be easily and simply deployed.<br>♦ It can effectively suppress the uncertainty disturbance. | 2000 | [153] |
| | SF | PI Controller | ♦ A periodic control strategy is proposed for the feed characteristics of the incinerator. | 2003 | [154] |
| | SF | Linear quadratic regulator | ♦ A full-state closed-loop feedback loop with inside loop is designed. | 2020 | [155] |
| MIMO | SF, and FGOC | Linear model predictive control | ♦ It can effectively suppress the influence of large disturbance on the control system. | 2005 | [26] |
| | SF, and FGOC | Nonlinear model predictive control | ♦ A moving horizon estimator is used to estimate the states and disturbances. | 2005 | [156] |
| | SF, and FGOC | PID controller | ♦ Add disturbance rejection loops to improve controller performance. | 2010 | [157] |
| | FT, and FGOC | Fuzzy neural network | ♦ The controller structure is self-organized and adjusted by calculating the similarity of neurons and multitask learning ability. | 2023 | [158] |
| | FT, SF, and FGOC | PID controller | ♦ A quasidiagonal recurrent neural network is used to adjust the control parameters automatically. | 2022 | [159] |
| | FT, SF, and FGOC | Single neuron adaptive PID controller | ♦ A multivariable serial control structure is designed based on the process flow. | 2023 | [160] |

## 5. AI Application Research in Optimization of MSWI Process

The optimization of key controlled variables (FT, FGOC, SF, CLP, etc.) in the MSWI process has received limited attention in the existing literature. Prevailing studies predominantly focus on the setpoints of manipulated variables associated with "air distribution" (AD) and "material distribution" (MD) [161]. Smart optimal control of the MSWI process necessitates the simultaneous minimization of exhaust emissions, material consumption, and combustion efficiency while optimizing other pertinent product indices. This must be achieved within the constraints of various equality and inequality constraints, necessitating the utilization of intelligent optimization algorithms to address multi-objective conflicts.

Within the realm of air distribution, Xia et al. [161] applied case-based reasoning (CBR) informed by domain expert knowledge to intelligently determine setpoints. In a similar vein, Ding et al. [162] intelligently adjusted the secondary air volume, achieving optimal settings. The core concept of CBR-based intelligent setting involves the reuse of expert experience, offering conformity to empirical cognition. However, it is limited in range, posing challenges in finding true optimal setpoints. Recent efforts to optimize primary/secondary air volume setpoints introduced a multi-objective PSO algorithm [43] and a multicondition operational optimization with adaptive knowledge transfer algorithm [27]. The algorithm, validated with industrial field data, demonstrated robust global optimization ability.

Concerning material distribution, Anderson et al. [49] employed a multi-objective evolutionary algorithm to ascertain optimal setpoints for feed rate, effectively achieving the goals of maximizing the feed rate and minimizing ash carbon content.

While these studies yield positive outcomes under single operating conditions, challenges persist, such as adaptability to multiple operating conditions, consideration of multiple objectives, and integration of multimodal data.

Studies addressing the optimization of controlled variables (CV) are limited. Huang et al. [163] utilized the multi-objective competitive swarm optimization algorithm to optimize furnace temperature and flue gas oxygen content, achieving multi-objective conflict optimization for NOx and combustion efficiency. However, this study focused solely on providing setpoints for key controlled variables without considering the controller, necessitating further exploration in this research area.

Optimal research in the MSWI process presents numerous challenging problems, including determining optimal setpoints for multiconflicting objectives, managing multiple controlled variables, and optimizing whole process operational indices using multimodal data-driven approaches.

The optimization research studies are summarized in Table 5.

**Table 5.** Summary of optimization research studies.

| Object | Technology | | Benefit | Year | Literature |
|---|---|---|---|---|---|
| AD | Case-based reasoning | ◆ | The effectiveness of the proposed method is verified in the simulation platform. | 2020 | [161] |
| | Case-based reasoning, random weight neuron network, and radial basis function | ◆ | The optimal setting of secondary air volume is realized by integrating multiple intelligent algorithms. | 2022 | [162] |
| | Multi-objective particle swarm optimization | ◆ | Based on the population state, the corresponding update methods are designed to improve the problem of falling into local optimum. | 2023 | [43] |
| | Multi-objective particle swarm optimization | ◆ | An adaptive knowledge transfer strategy is designed to improve optimization efficiency. | 2023 | [27] |
| MD | Multi-objective genetic algorithm | ◆ ◆ | FLIC is used to generate model training data. It has strong expansibility and portability. | 2005 | [49] |
| CV | Multi-objective competitive swarm optimization | ◆ ◆ | A comprehensive nondominated evaluation system and improved competitive mechanism are proposed. An adaptive scheme combined with multistrategy learning is proposed. | 2024 | [163] |

## 6. AI Application Research in Maintenance of MSWI Process

Domain experts rely on DCS monitoring of process data, industrial camera observations of furnace flames, and on-site information gathered through regular manual inspections to diagnose faults in MSWI plants. However, several persistent problems hinder this diagnostic process:

(1) The information within the DCS system undergoes frequent changes. The alarm function for abnormal operating conditions is solely triggered based on whether the collected data exceed a limit value, resulting in false alarms and complicating issue tracing.

(2) The high temperature and intense light during the combustion process, coupled with molten material production, impede the industrial camera's ability to capture a clear flame picture. This poses challenges for operating engineers in making informed decisions, potentially leading to fluctuations in operating conditions.

(3) In high-temperature and noisy environments, inspection engineers can only assess the normality of equipment by listening, posing challenges in ensuring optimal operation.

The fault diagnosis mode conducted by domain experts faces challenges such as suboptimality, delay, and subjectivity, making it difficult to ensure the safety, stability, and optimal operation of the MSWI process under these circumstances.

### 6.1. Recognition of Flame Status

Flame features, encompassing the combustion line position and essential physical attributes such as area, height, and brightness, play a pivotal role in evaluating the flame status within the combustion process [164]. These features directly influence the comprehension of phenomena like partial burning, local burning through, coking, ash deposition, and corrosion occurring in the furnace [165]. Researchers have devised various approaches for recognizing the combustion status based on the combustion line position and other flame characteristics. Duan et al. [39] employed a combustion status recognition model that integrates multiscale color moment features and the RF algorithm. Guo et al. [72] introduced a strategy for combustion status recognition relying on the hybrid enhancement of generative adversarial networks (GANs). Pan et al. [166] proposed an innovative online recognition method using deep forest classification (DFC) based on convolutional multi-layer feature fusion. The strategy diagram for Pan et al.'s method is illustrated in Figure 11.

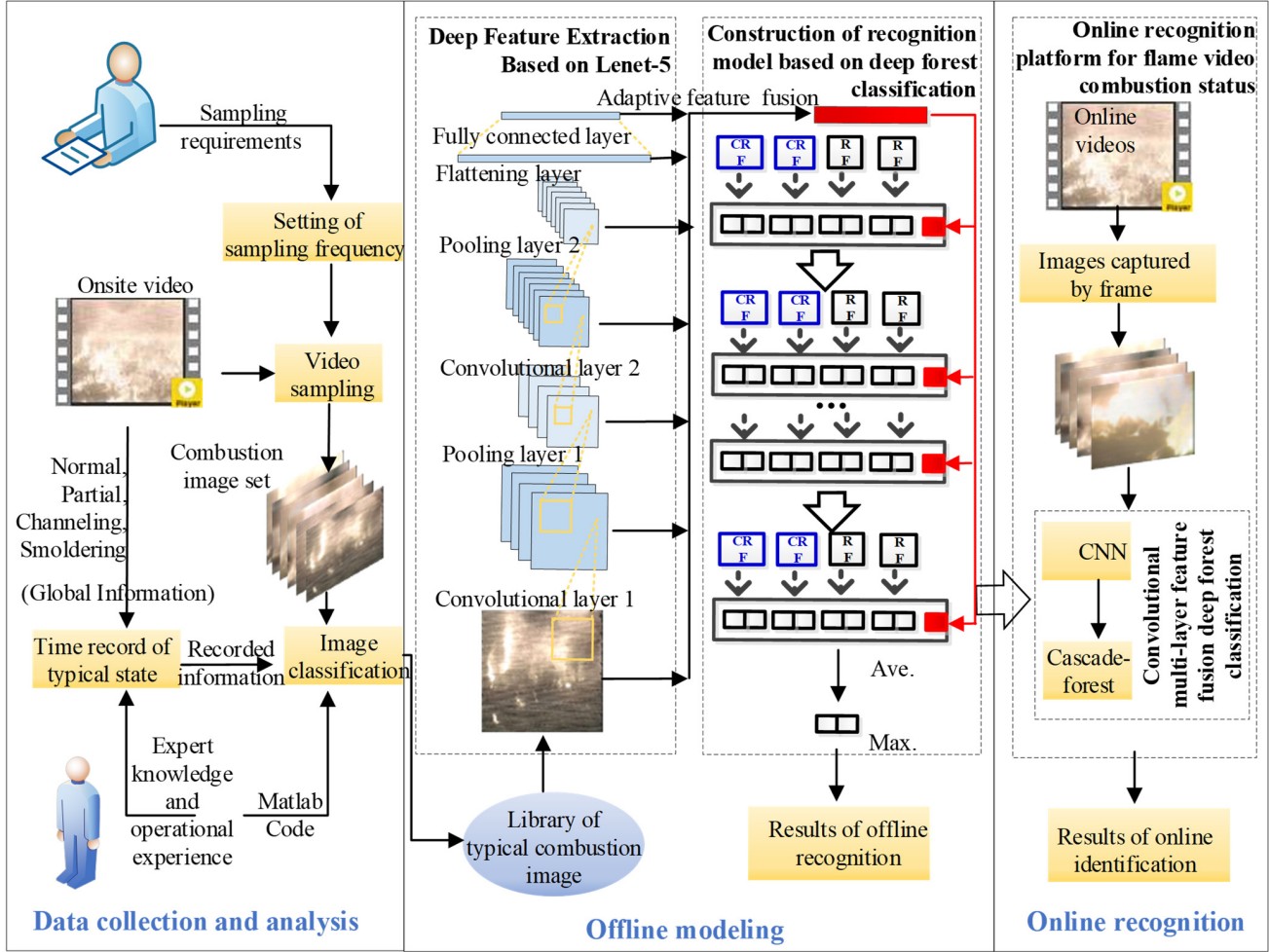

**Figure 11.** Strategy diagram of Ref. [166].

Han et al. [167] proposed a recognition model for unknown flame combustion status using a semisupervised approach. Sun et al. [168] aimed to uncover the relationship between combustion flame images and temperature, reconstructing the temperature field in each region of the flame using the acoustic emission temperature detection method. Zheng et al. [169] combined the Newton iteration method and Hottel emissivity model to establish a relationship model between multispectral flame images and temperature. Yan [170] and He [171] et al. employed spectrometers to detect the flame and constructed mapping models between its characteristics and the concentration of alkaline metals emit-

ted. Zhou et al. [172] conducted three-dimensional visualization modeling of flame temperature based on Monte Carlo and multi-imaging angles. Notably, most of these studies relied on additional physical equipment and did not consider the recognition of combustion status based on multimodal data.

In summary, there is a pressing need for further research on detecting combustion status based on multimodal data and performing fusion verification.

### 6.2. Qualitative Detection of Operational Fault

Since the 1990s, researchers have delved into the application of computer and AI technology in the fault diagnosis of the MSWI process to aid domain experts in decision making. Ono et al. [61] developed a fuzzy expert fault reasoning system for the incineration and boiler system, capable of conducting symptom analysis, early warning, fault alarm, analysis, and recognition within and outside the limits of the DCS system. Subsequently, Chen et al. [173] performed online diagnosis of abnormal exhaust emission and steam flow using cluster analysis, neural networks, and Monte Carlo statistics. Addressing issues like MSW partial burning, through poor slag discharge, and coking in the furnace, Tao et al. [174] constructed a fault tree based on process analysis and historical experience. They employed a rule reasoning expert system for detection, achieving an experimentally validated accuracy rate of 90%. Concurrently, Tao et al. [44] employed a BPNN modeling strategy for diagnosis. For the recognition of combustion status, Zhou et al. [53] established a BPNN-based diagnostic model with a high accuracy rate of 99%. However, challenges such as overfitting and high requirements for training samples were noted. Additionally, Ding et al. [74] built a CBR model based on RWNN similarity retrieval for diagnosing superheater and economizer leakage, ash deposition, slagging in horizontal flues, as well as coking and poor slag discharge in the furnace. Experimental results demonstrated satisfactory performance. However, these studies primarily focus on constructing classifier models capable of determining whether a fault occurs, lacking quantification or localization of the fault.

### 6.3. Quantitative Detection of Operational Fault

Multivariate statistical process monitoring (MSPM) technology, leveraging industrial data for quantitative fault detection, has garnered extensive attention from both industry and academia [175–177]. The basic strategy involves establishing a latent structure model using normal operating condition data. Subsequently, high-dimensional variables are projected into a low-dimensional space, and statistical indices such as squared prediction and Hotelling's $T^2$ are compared to determine whether a fault occurs. Finally, fault location is performed through data reconstruction. Zhao et al. [64] were among the pioneers in employing PCA and rule reasoning for the quantitative detection of incinerator faults, demonstrating a significant reduction in the false positive rate associated with faults. Similarly, Tavares et al. [28] conducted a comparative analysis between PCA-based and PLS-based fault diagnosis approaches. The results not only indicated superior overall performance but also highlighted the effectiveness of various statistical indices in precisely locating faults. It is noteworthy that the existing literature on fault quantitative detection in MSWI processes is limited and predominantly relies on the linear PCA/PLS method. The dynamic, nonlinear, multiscale, and multimodal characteristics inherent in the MSWI process present novel challenges to both theoretical and applied research in MSPM.

The maintenance research studies are summarized in Table 6.

Table 6. Summary of maintenance research studies.

| Object | Technology | | Benefit | Year | Literature |
|---|---|---|---|---|---|
| Recognition of flame status | Multiscale color moment features and random forest | ◆ ◆ | Certain interpretability. Combining local and global features. | 2019 | [39] |
| | Generative adversarial network | ◆ | Increasing the number of modeling samples. | 2022 | [72] |
| | DFC based on convolutional multilayer feature fusion | ◆ ◆ | Developing an online combustion status-recognition platform. Integrating deep fusion features with the DFC. | 2023 | [166] |
| Qualitative detection of operational fault | Fuzzy expert system | ◆ ◆ | Good visualization. Strong practicality. | 1994 | [61] |
| | Cluster analysis, artificial neural networks, and Monte Carlo simulation | ◆ | Accurate state monitoring for steam generation and NOx control. | 2008 | [173] |
| | Fault tree and expert system | ◆ | Continued updates. | 2008 | [174] |
| | Back propagation neural network | ◆ | Integrating multiple neural networks. | 2008 | [44] |
| | Back propagation neural network | ◆ | High accuracy. | 2015 | [53] |
| | Radom weight neuro network and case-based reasoning | ◆ | Reduction in the detection time complexity. | 2021 | [74] |
| Quantitative detection of operational fault | Principal component analysis | ◆ ◆ | Earlier than human operators. Reducing the misreporting rate. | 2008 | [64] |
| | Principal component analysis and partial least square | ◆ | Good performance in fault detection and isolation. | 2011 | [28] |

## 7. Outlook on AI Application for MSWI Process

The integration of AI with a specific industrial domain is referred to as industrial AI technology. At its core, this technology aims to facilitate innovative applications such as high-performance controllers, intelligent operational decision making, and intelligent algorithm updating [3]. Its primary objective is to seamlessly adapt to the intricate and dynamic industrial environment to accomplish diverse operational goals and tasks [178]. Achieving sustainable development in the MSWI process necessitates a profound integration with industrial AI technology, thereby diminishing reliance on domain experts. Urgent attention is required for research in AI technology on modeling operational indices, intelligent control of the combustion process, collaborative optimization of the whole process, and intelligent maintenance of the overall system.

### 7.1. Operational Indices Modeling

The effective operation of an integrated system facilitating intelligent optimization, decision making, and control for complex industrial processes hinges on the ability to conduct real-time monitoring of key operational indices [179]. Clearly, the online detection of environmental, product, and economic indices plays a pivotal role in ensuring the safe, stable, and optimal operation of the MSWI process. The modeling and prediction of various industrial processes [180] can be accomplished by constructing intelligent models based on easily collectible multimodal data, such as process variables and flame videos. The distinctive characteristics of the MSWI process, including its multiprocess, multistage nature, complexity, and unclear mechanisms, result in different time scales, variations, and uncertainties in modeling samples. It is imperative to analyze the delay characteristics

between process variables and operational indices based on the mechanisms of thermal power transmission and chemical substance conversion, as well as the correlation between multitemporal and spatial scale samples and operational indices. The process variables within the MSWI process are numerous and intricately coupled. The correlation between operational indices and various process stages varies, with an unclear underlying mechanism. Utilizing numerical simulation becomes imperative to elucidate these mechanisms, drawing upon the expertise of domain professionals and extracting knowledge embedded in data to minimize model input. In addressing high-dimensional sparse modeling samples, techniques like virtual sample generation are employed to expand sample numbers, mitigating issues related to imbalanced sample distribution and unknown expected distribution. A study on AI algorithms with strong interpretability is essential for modeling and predicting operational indices. The actual MSWI process exhibits numerous interference factors and frequent fluctuations in operating conditions, necessitating an operational index model capable of adaptive adjustments to dynamic process changes for accurate predictions. Employing methods based on mathematical models [181,182], multivariate statistics [183,184], and AI [185,186] aids in predicting drift time, drift degree, and drift position under new operating conditions specific to MSWI operational indices. Incorporating adaptive update algorithms, continuous learning mechanisms, knowledge transfer, and incremental learning strategies becomes important to enhance the robustness and generalization performance of online modeling.

In recent years, researchers have conducted numerous studies on modeling operational indices in complex industrial processes, such as blast furnace ironmaking, fused magnesium, and petrochemical processes. Addressing the challenge of sparse labeled samples for modeling, various methods have been proposed, including virtual sample generation [187,188], semisupervised [189], weakly supervised [190], and unsupervised [191]. These strategies offer robust support for investigating the sample completion mechanism in the modeling of the MSWI process. To enhance multisource information representation and model interpretability, diverse methods have been introduced, including multifeature information fusion [192], multimodal deep learning [193], visual data depth modeling [194], Bayesian data-driven T-S fuzzy [195], and deep forest regression [66,196]. These serve as the theoretical foundation for exploring intelligent reduction in multisource features and constructing interpretable models in the MSWI process. Confronting the challenge of online dynamic prediction, studies on broad learning systems [197,198], concept drift learning [199], and dynamic self-organization model [200] indirectly demonstrate the promising feasibility of developing intelligent prediction systems for operational indices.

### 7.2. Intelligent Control of Combustion Process

The combustion process in MSWI faces notable challenges in achieving precise control compared to power generation processes utilizing coal and gas as input materials, primarily due to numerous interference factors and frequent fluctuations in operating conditions [101,159]. Despite the abundance of process data, their distribution is unbalanced. Additionally, the integration of unstructured data, such as images and videos, with process data is hindered by time delays and information asymmetry, presenting challenges in fusion. Furthermore, obtaining, quantifying, and utilizing knowledge related to the incineration mechanism poses difficulties. The MSWI process exhibits significant nonstationary characteristics, including frequent transitions between steady-state conditions and transition conditions, as well as substantial sensor drift in high-temperature and high-pressure environments, among other factors. These factors underscore the importance of effective operating condition perception and fault diagnosis for each process stage as crucial guarantees to ensure the stable operation of the controller. Additionally, the development of models for operating condition perception and fault diagnosis encounters challenges such as a lack of samples, unknown types, difficulty in explanation, uncertain changes/occurrences, and unidentified potential faults. Given the regional and seasonal variations in MSW composition and calorific value, coupled with the diverse experiences of operators and varying

levels of operation and maintenance, the MSWI process exhibits a multitude of operating conditions. Traditional PID controllers prove inadequate in adapting to these different situations. Effective perception of operating conditions necessitates the adoption of diverse intelligent control algorithms tailored to different operating conditions. Research indicates that intermittent operation in the MSWI process results in severe pollution, a significant increase in operating costs, and challenges in processing capacity [201]. Consequently, a controller operating under dynamic disturbance must possess the capability for fault tolerance, robustness, and adaptability.

The rapid advancement of AI technology has spurred numerous studies on intelligent control in various industrial processes. For instance, studies have explored adaptive sliding mode control [202], fuzzy neural control [203], and reinforcement-learning-based tracking control [204] for wastewater treatment processes. Event-triggered control [205], adaptive tracking control [206], and model predictive control [207] have been investigated for continuous stirred-tank reactor systems. Similarly, model-free adaptive predictive control [208], model predictive control [209], and fuzzy control [210] have been applied to blast furnace ironmaking process. These studies not only establish a theoretical foundation but also provide technical support for the controlled object model and basic loop intelligent controller in the MSWI process. Furthermore, research on data-driven operating condition monitoring in coal-fired power generation processes [211] and self-organizing control in wastewater treatment process [202] and blast furnace ironmaking process [212] contributes to the exploration of intelligent condition perception, fault diagnosis, and the self-organizing mechanism of intelligent controllers under dynamic conditions.

### 7.3. Collaborative Optimization of Whole Process

In the field of collaborative optimization of industrial processes, a category of challenges arises, involving mixed, multi-objective, multiconstraint, and multiscale dynamic conflict optimization problems [213,214]. The intelligent optimal decision making in human–machine collaboration encompasses tasks such as feed selection, operation and maintenance, and on-site decisions. The collaborative optimization of the whole process assumes that each process stage's control system in the MSWI process acts as an independent agent, and setpoints are collaboratively determined with the aim of optimizing multiple conflicting and multiscale operational indices. The selection and configuration of process parameters, as well as the optimal operation of the whole process, primarily depend on domain experts in the actual industrial setting. Nevertheless, the qualitative expression of operational status and decision making information encounters challenges related to inaccuracy, uncertainty, fuzziness, and even nonuniqueness. Optimal operation stands as the core of intelligent control [215,216], involving the resolution of real operational statuses or process planning problems through optimization. Addressing the actual needs of an MSWI plant, achieving the optimal solution for multiconflict objectives holds great significance in realizing smart optimal control [49]. The environmental, product, and economic indices in MSWI processes exhibit characteristics of multiconflict, multiconstraint, dynamic time-varying, and multispatial scales, given the limitations of processes and technologies. Consequently, the optimization problem in the MSWI process can be conceptualized as a multi-objective function extremum problem under multiconstraint conditions, representing a notably challenging issue. The diverse sources and complex components of MSW, coupled with the variability and fluctuations in operating conditions, further intensify the complexity of real-time optimal operation. In the context of intelligent optimization in the process industry, Chai et al. [217,218] highlighted that human–machine collaboration and interactive learning between domain experts and intelligent optimal decision-making systems are pivotal directions for future development. The MSWI process aspires to achieve minimal energy and material consumption, zero emission of pollutants, and environmental greening. This necessitates the capabilities of perception, cognition, decision making, and execution possessed by domain experts. In nature, achieving intelligent decision making

in complex industrial processes relies on the enhanced interactive evolution of domain experts and industrial AI technology.

Collaborative optimization decision making, integrating new AI technology, emerges as a feasible approach to achieve intelligent operation and reduce reliance on domain experts in industrial processes [219]. Numerous studies in this domain have yielded outstanding cases that serve as valuable references. Noteworthy achievements in collaborative optimization include distributed optimization subject to inseparable coupled constraints in the ethylene process [220], multi-objective optimization under the dynamic environment of the iron removal process [221], carbon emission optimization under different time scales in the sintering process [222], blast furnace charge surface optimization with feedback compensation [223], and intelligent optimization of the ironmaking process supported by a mixed model [224,225]. These accomplishments provide valuable guidance and support for the collaborative optimization of the entire process in the MSWI process. In the field of intelligent optimization decision making, studies on the intelligent decision making of the entire process operation in mineral processing [226], scheduling optimization of the ethylene cracking furnace system [227], multifurnace optimal scheduling of silicon single crystal and fused magnesium production processes [228,229], and others have been successfully conducted. These achievements underscore that research on multi-objective real-time optimization algorithms and human–machine collaborative enhanced interactive evolution for intelligent decision making in the MSWI process has a solid theoretical foundation and practical feasibility.

### 7.4. Intelligent Maintenance of Whole Process

The enhanced automation of the MSWI process brings about a considerable increase in both complexity and uncertainty, subsequently raising the possibility and severity of faults [101]. Failure to detect and address faults promptly may result in substantial losses. However, manual monitoring faces challenges, particularly in identifying minor faults that are difficult to detect. Hence, intelligent maintenance for the MSWI process becomes imperative.

In practice, the MSWI process undergoes transitions between diverse operating conditions based on factors such as waste components, feeding amounts, combustion temperature, and changes in setpoint values, aligning with the requirements of production indicators and safety standards. Consequently, addressing the quantification and evaluation of MSWI process operating conditions is a prominent challenge. Distinct operating conditions exert varying influences on system stability, efficiency, and pollutant emissions. Given the extended duration, multistage structure, and multifaceted nature of the MSWI process, the requirements at each operational stage lack consistency. The characteristics of process data undergo rapid and frequent changes with the shifting operational stages, posing tracking difficulties. Furthermore, the protracted operation cycle of the MSWI process results in extended device runtimes, impacting the current condition due to the influence of previous operational conditions. Therefore, swift and accurate recognition of multistage operational conditions stands as a key aspect of MSWI process monitoring, necessitating consideration of time continuity. In comparison to normal conditions, abnormal occurrences are less frequent and may be singular. Consequently, there exists a substantial difference in the number of process data instances between normal and abnormal operating conditions, giving rise to a class imbalance problem [230,231]. Constructing an accurate and robust fault diagnosis model based on class imbalance data proves challenging, leading to considerable instances of false positives and false negatives.

Process monitoring and fault diagnosis, driven by AI technologies, have found widespread application in complex industrial processes, including petrochemicals, metallurgy, and energy. Notably, to address condition-switching challenges in multimode process monitoring, various techniques have been proposed. Examples include the hybrid cluster variational autoencoder designed for blast furnace ironmaking [232], similarity-preserving dictionary learning applied to the roasting process [233], and dynamic locality-preserving

PCA tailored for power generation processes [234]. These approaches aim to solve diverse condition-switching problems encountered in process monitoring. Existing research on complex industrial processes demonstrates the feasibility of multimode MSWI process monitoring. To address class imbalance problems in fault diagnosis of mechanical equipment, widely practiced techniques include virtual sample generation (VSG) and transfer learning. Illustrative instances encompass the features selection oversampling technique for bearing fault diagnosis [235], SMOTEBoost for rotor-bearing systems [236], and adversarial transfer learning applied to planetary gearboxes [237]. These approaches provide valuable references and support for tackling class imbalance challenges in MSWI fault diagnosis.

## 8. Conclusions

In conclusion, significant research has been conducted on the application of AI technology to the MSWI process. This study systematically reviews AI research of modeling, control, optimization, and maintenance, addressing the foundational challenges associated with optimal control. AI technology plays an essential role in fostering the ongoing development of the MSWI process. However, there is a discernible widening gap between academic research and industrial application. To effectively bridge this gap, future research should pay attention to the following aspects: (1) establish a dynamic, robust, and interpretable MSWI process model with AI for both controlled variables and operational indices; (2) construct a steady-state AI-based loop controller tailored for diverse operational conditions, along with its self-organizing mechanism under strong dynamic interference; (3) address issues such as the integration of on-site data and off-site knowledge with AI and the implementation of a dynamic multiscale multi-objective optimization algorithm; (4) develop fast and accurate drift monitoring of complex operating conditions, and fault detection technology based on few and zero samples in terms of multimodal data. Therefore, this study addresses the lack of a review on AI application in the field of WTE, especially MSWI, which provides clarity for future research. It is believed that AI will play a more significant role in the optimal control of the MSWI process.

**Author Contributions:** Methodology, J.T.; Writing—original draft, T.W.; Resources, Data curation, H.X.; Writing—review and editing, C.C. All authors have read and agreed to the published version of the manuscript.

**Funding:** This research received no external funding.

**Institutional Review Board Statement:** Not applicable.

**Informed Consent Statement:** Not applicable.

**Data Availability Statement:** The data presented in this study are available on request from the corresponding author.

**Conflicts of Interest:** The authors declare that they have no known competing financial interests or personal relationships that could have appeared to influence the work reported in this article.

## Abbreviations

| Abbreviations | Meanings |
| --- | --- |
| AI | Artificial intelligence |
| IoT | Internet of things |
| MSW | Municipal solid waste |
| MSWI | Municipal solid waste incineration |
| WTE | Waste-to-energy |
| WoS | Web of Science |
| CNKI | China National Knowledge Internet |
| SNCR | Selective noncatalytic reduction |
| PSO | Particle swarm optimization |
| PCA | Principal component analysis |
| PLS | Partial least squares |

| | |
|---|---|
| NN | Neural network |
| RBFNN | Radial basis function neural network |
| MNN | Modular neural network |
| LS-SVM | Least square-support vector machine |
| DBN | Deep belief network |
| DFR-clfc | Deep forest regression based on cross-layer full connection |
| IDFR | Improved deep forest regression |
| SVM | Support vector machine |
| TM | Tree-based model |
| FL | Fuzzy logic |
| FNN | Fuzzy neural network |
| DL | Deep learning |
| FT | Furnace temperature |
| FGOC | Flue gas oxygen content |
| SF | Steam flow |
| CLP | Combustion line position |
| MISO | Multi-input single-output |
| LS-SVR | Least squares-support vector regression |
| LSTM | Long short-term memory network |
| RBF | Radial basis function |
| MIMO | Multi-input multi-output |
| RF | Random forest |
| GBDT | Gradient boost decision tree |
| CVMSW | Calorific value of municipal solid waste |
| ANFIS | Adaptive network based fuzzy inference system |
| ANN | Artificial neural network |
| TMSWL | Thickness of the municipal solid waste layer |
| CEMS | Continuous emission monitoring system |
| DXN | Dioxin |
| VOCs | Volatile organic compounds |
| CO | Carbon monoxide |
| BPNN | Back propagation neural network |
| SVR | Support vector regression |
| APCDs | Air pollution control devices |
| HRR | HRR |
| ACC | Automatic combustion control |
| SISO | Single-input and single-output |
| HSIC | Human simulated intelligent controller |
| LMPC | Linear model predictive control |
| NMPC | Nonlinear model predictive control |
| PID | Proportional integral differential |
| DCS | Distributed control system |
| AD | Air distribution |
| MD | Material distribution |
| CBR | Case-based reasoning |
| CV | Controlled variables |
| GANs | Generative adversarial networks |
| DFC | Deep forest classification |
| CBR | Case-based reasoning |
| RWNN | Random weight neural network |
| MSPM | Multivariate statistical process monitoring |
| VSG | Virtual sample generation |

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
