# Peer review of "An Overview of Artificial Intelligence Application for Optimal Control of Municipal Solid Waste Incineration Process"

_sustainability, doi:10.3390/su16052042_

Round 1

Reviewer 1 Report

Comments and Suggestions for Authors

Dear Authors,

Thank you for interesting and comprehensive research. However, I have a few minor suggestions to improve your paper.

1. Could you please describe the limitations of your research? 

2. Could you give at least a short description or a few ideas of needed future research in the field?

3. How does your paper contribute to the research field of MSWI and WTE. I suggest emphasizing a few strongest points regarding this.

Author Response

Point 1: Could you please describe the limitations of your research?

Response 1:

Thanks.

In this study, the researches on the MSWI process are reviewed in terms of optimal control in terms of four prospects such as modeling, control, optimization, and maintenance. One of the limitations is that the too many references are reviewed with limited details. That is to say, each prospect such as modeling, control, optimization, and maintenance should be viewed with detailed AI algorithm and their applications, which can be more useful for the researches of these sub-domain. Another limitation is that the first principal model are not be reviewed. In the future, we will focus on these limitations.

Point 2: Could you give at least a short description or a few ideas of needed future research in the field?

Response 2:

Thanks.

After reviewing the existing studies, we have given the focus of future research in conclusion, as follows:

In conclusion, significant research has been conducted on the application of AI technology to the MSWI process. This study systematically reviews AI research of modeling, control, optimization, and maintenance, addressing the foundational challenges associated with optimal control. AI technology plays an essential role in fostering the ongoing development of the MSWI process. However, there is a discernible widening gap between academic research and industrial application. To effectively bridge this gap, future research should pay attention to the following aspects: 1) establish a dynamic, robust, and interpretable MSWI process model with AI for both controlled variables and operational indices; 2) construct a steady-state AI-based loop controller tailored for diverse operational conditions, along with its self-organizing mechanism under strong dynamic interference; 3)address issues such as the integration of on-site data and off-site knowledge with AI and the implementation of a dynamic multi-scale multi-objective optimization algorithm; 4)develop fast and accurate drift monitoring of complex operating conditions, and fault detection technology based on few and zero samples in terms of multi-modal data. Therefore, this study addresses the lack of a review on AI application in the field of WTE, especially MSWI, which provides clarity for future research. It is believed that AI will play a more significant role in the optimal control of the MSWI process.

Point 3: How does your paper contribute to the research field of MSWI and WTE. I suggest emphasizing a few strongest points regarding this.

Response 3:

Thanks.

We have rewritten the abstract section to highlight the contributions of this article, as follows:

Artificial intelligence (AI) has found widespread application across diverse domains, including residential life and product manufacturing. Municipal solid waste incineration (MSWI) represents a significant avenue for realizing waste-to-energy (WTE) objectives, emphasizing resource reuse and sustainability. Theoretically, AI holds the potential to facilitate optimal control of the MSWI process in terms of achieving minimal pollution emissions and maximal energy efficiency. However, a noticeable shortage exists in the current research of review literature concerning AI in the field of WTE, particularly MSWI, hindering a focused understanding of future development directions. Consequently, this study conducts an exhaustive survey of AI applications for optimal control, categorizing them into four fundamental aspects: modeling, control, optimization, and maintenance. Timeline diagrams depicting the evolution of AI technologies in the MSWI process are presented to offer an intuitive visual representation. Each category undergoes meticulous classification and description, elucidating the shortcomings and challenges inherent in current research. Furthermore, the study articulates the future development trajectory of AI applications within the four fundamental categories, underscoring the contribution it makes to the field of MSWI and WTE.

Reviewer 2 Report

Comments and Suggestions for Authors

The authors have summarized the researches on the application of artificial intelligence on the control of municipal solid waste incineration process. Overall, it is an interesting and meaningful work. The topic is novel, the structure is complete and the summary is relatively comprehensive. Some suggestions are listed below to discuss with the authors.

1. Line 483-508. The control in on-site or the close-loop control strategy of MSWI is important for MSWI industry in the future. The economic-technic potential or costs of utilizing this control strategy are encouraged to be summarized if possible.

2. The conclusion part is suggested to be more concise to emphasize its novelty.

Author Response

Point 1: Line 483-508. The control in on-site or the close-loop control strategy of MSWI is important for MSWI industry in the future. The economic-technic potential or costs of utilizing this control strategy are encouraged to be summarized if possible.

Response 1: Thanks.

We have added a description in Section 4, as follows:

These studies underscore the divergence in research focus between industrial ap-plications and academic research. Clearly, employing AI for on-site or closed-loop control of the MSWI process not only addresses the issue of operating condition fluctuations arising from manual control modes and the expertise of domain experts but also contributes to cost reduction, enhanced energy efficiency, and reduced pollution emissions. Consequently, AI-assisted control is poised to become the prevailing trend in future research. However, the current AI algorithms researched face challenges in direct application due to the closure of the DCS system and safety requirements within MSWI enterprises. Therefore, the establishment of a hardware-in-loop simulation experimental platform is deemed necessary for testing and validating AI algorithms in industrial settings [117]. Clearly, the challenge lies in conducting more in-depth academic research in terms of application to enhance its universality and applicability to the actual industry, posing a complex problem that requires further exploration.

Point 2: The conclusion part is suggested to be more concise to emphasize its novelty.。

Response 2:

Thanks.

We have revised the conclusion, as follows:

In conclusion, significant research has been conducted on the application of AI technology to the MSWI process. This study systematically reviews AI research of modeling, control, optimization, and maintenance, addressing the foundational challenges associated with optimal control. AI technology plays an essential role in fostering the ongoing development of the MSWI process. However, there is a discernible widening gap between academic research and industrial application. To effectively bridge this gap, future research should pay attention to the following aspects: 1) establish a dynamic, robust, and interpretable MSWI process model with AI for both controlled variables and operational indices; 2) construct a steady-state AI-based loop controller tailored for diverse operational conditions, along with its self-organizing mechanism under strong dynamic interference; 3)address issues such as the integration of on-site data and off-site knowledge with AI and the implementation of a dynamic multi-scale multi-objective optimization algorithm; 4)develop fast and accurate drift monitoring of complex operating conditions, and fault detection technology based on few and zero samples in terms of multi-modal data. Therefore, this study addresses the lack of a review on AI application in the field of WTE, especially MSWI, which provides clarity for future research. It is believed that AI will play a more significant role in the optimal control of the MSWI process.

Reviewer 3 Report

Comments and Suggestions for Authors

The manuscript provides an overview of AI applications in the municipal solid waste incineration process.
The manuscript is overall well written. While comprehensive, the manuscript is not too extensive. Figures are adequate and aid in a better understanding of the process. An adequate number of recent and relevant studies have been addressed. Minor editing of English is required; some sentences can be better rephrased. 

Comments on the Quality of English Language

Minor editing of English is required; some sentences can be better rephrased. 

Author Response

Point 1: The manuscript is overall well written. While comprehensive, the manuscript is not too extensive. Figures are adequate and aid in a better understanding of the process. An adequate number of recent and relevant studies have been addressed. Minor editing of English is required; some sentences can be better rephrased.
Response 1: 
Thanks. 
We have added a figure in the Introduction section of the revised revision to clearly show the structure of this study, as follows:
The structure of this study is shown in Figure 2.

Figure 2. Structure of this study
Section 2 introduces the literature review methodology, provides a detailed description of the MSWI process, and offers a brief overview of AI applications for optimal control. Subsequently, sections 3-6 delve into the individual fields of AI application research, addressing modeling, control, optimization, and maintenance of the MSWI process. In Section 7, the focus shifts to an in-depth discussion of the prospects and outlook on AI applications within the MSWI process. Finally, Section 8 encapsulates the key findings and conclusions of this study.

Reviewer 4 Report

Comments and Suggestions for Authors

1-   This review article investigated the application of AI technology to the MSWI process and systematically examines AI research relating to modeling, control, optimization, and maintenance. Once again, this review article confirms the huge gap between industry and academia.

2-   The title reflects the content of the paper.

3-   The abstract section gives the summary of the work.

4-   The research originality, knowledge gap, problem statement and research significance of the present study are clearly explained.

5-   The introduction and literature review has been explained well.

6-   The methodology is straightforward and its description seems adequate for the purpose of the work.

7-   The tables and figures clearly and easily show data visually and easy to understand.

8-   The results are well presented

9-   The conclusions perform the findings of the present study and it seems consistent with the results; future research endeavors focusing on AI applications relating to aspects of modeling, control, optimization, and maintenance have been recommended.

The article adequately referenced.

Author Response

Point 1: This review article investigated the application of AI technology to the MSWI process and systematically examines AI research relating to modeling, control, optimization, and maintenance. Once again, this review article confirms the huge gap between industry and academia.

Point 2: The title reflects the content of the paper.

Point 3: The abstract section gives the summary of the work.

Point 4: The research originality, knowledge gap, problem statement and research significance of the present study are clearly explained.

Point 5: The introduction and literature review has been explained well.

Point 6: The methodology is straightforward and its description seems adequate for the purpose of the work.

Point 7: The tables and figures clearly and easily show data visually and easy to understand.

Point 8: The results are well presented.

Point 9: The conclusions perform the findings of the present study and it seems consistent with the results; future research endeavors focusing on AI applications relating to aspects of modeling, control, optimization, and maintenance have been recommended.

The article adequately referenced.

Response:

Thanks.

We have further improved the structure and content of the article, and further summarized the literature. At the same time, the abstract has been modified to highlight the contribution of this study, as follows:

Artificial intelligence (AI) has found widespread application across diverse domains, including residential life and product manufacturing. Municipal solid waste incineration (MSWI) represents a significant avenue for realizing waste-to-energy (WTE) objectives, emphasizing resource reuse and sustainability. Theoretically, AI holds the potential to facilitate optimal control of the MSWI process in terms of achieving minimal pollution emissions and maximal energy efficiency. However, a noticeable shortage exists in the current research of review literature concerning AI in the field of WTE, particularly MSWI, hindering a focused understanding of future development directions. Consequently, this study conducts an exhaustive survey of AI applications for optimal control, categorizing them into four fundamental aspects: modeling, control, optimization, and maintenance. Timeline diagrams depicting the evolution of AI technologies in the MSWI process are presented to offer an intuitive visual representation. Each category undergoes meticulous classification and description, elucidating the shortcomings and challenges inherent in current research. Furthermore, the study articulates the future development trajectory of AI applications within the four fundamental categories, underscoring the contribution it makes to the field of MSWI and WTE.

Reviewer 5 Report

Comments and Suggestions for Authors

The paper, unfortunately, does not meet the necessary criteria for an article that can be published in a serious journal, and even more so for a WoS-indexed journal. Below are some of my pointed observations. However, since the authors have conducted extensive literature research, I recommend redoing the article and submitting it for review. 

1.    Please review the abstract. Now it is very vague. I suggest revealing the main trends, and emphasizing the most important and catchy things from your literature review.

2.  Could you add a few lines about methodology how have you carried the literature review. Could you explain why you chose such structure of the paper? Maybe even a figure could help to reveal the logic of your literature review.

3. there is a complete lack of methodology explanation.

4. there is no quantification of the described statements and no real comparison between the cited studies.

5.  many concepts are stated with few references’ multiple times, even if the list of references is appropriate.

6. Figures from previous studies should be cited.

6. I understand the effort before a paper submission, and I regret to write that I do not see enough novelty and deep analysis compared to the long reference list. I would focus more on the review method and the literature comparison.

Author Response

Point 1: Please review the abstract. Now it is very vague. I suggest revealing the main trends, and emphasizing the most important and catchy things from your literature review.

Response 1:

Thanks.

We have rewritten the abstract to clearly highlight the focus and innovation of this study. The modified version is as follows:

Artificial intelligence (AI) has found widespread application across diverse domains, including residential life and product manufacturing. Municipal solid waste incineration (MSWI) represents a significant avenue for realizing waste-to-energy (WTE) objectives, emphasizing resource reuse and sustainability. Theoretically, AI holds the potential to facilitate optimal control of the MSWI process in terms of achieving minimal pollution emissions and maximal energy efficiency. However, a noticeable shortage exists in the current research of review literature concerning AI in the field of WTE, particularly MSWI, hindering a focused understanding of future development directions. Consequently, this study conducts an exhaustive survey of AI applications for optimal control, categorizing them into four fundamental aspects: modeling, control, optimization, and maintenance. Timeline diagrams depicting the evolution of AI technologies in the MSWI process are presented to offer an intuitive visual representation. Each category undergoes meticulous classification and description, elucidating the shortcomings and challenges inherent in current research. Furthermore, the study articulates the future development trajectory of AI applications within the four fundamental categories, underscoring the contribution it makes to the field of MSWI and WTE.

Point 2: Could you add a few lines about methodology how have you carried the literature review. Could you explain why you chose such structure of the paper? Maybe even a figure could help to reveal the logic of your literature review.

Response 2:

Thanks.

To express clearly, we have answered your questions point by point. They are shown as follows.

2.1 Could you add a few lines about methodology how have you carried the literature review.

Response 2.1:

In the revised version, we have adjusted the structure of the Section 2 by adding the relevant methodology description, as follows

2.1. Methodology about literature review

The literature reviewed in this study was systematically collected and processed from prominent scientific research databases, including WoS, Engineering Village, PubMed, and China National Knowledge Internet (CNKI). The retrieval time range spans from the establishment of each database to December 2023. Subsequently, the collected literature underwent a meticulous filtering process to exclude unrelated works. The refined literature was then categorized into four distinct sections based on AI applications in the MSWI process, namely modeling, control, optimization, and maintenance. Figure 3 illustrates the distribution of the literature across these four categories, providing an overview of the research landscape.

Figure 3. The number of literature in each category

Figure 3 reveals a notable concentration of research in the areas of modeling and control, while comparatively fewer studies focus on the optimization of the MSWI process using AI technology. It is crucial to highlight that optimization plays a pivotal role in achieving the sustainable development of the MSWI process. Despite the current emphasis on modeling and control, future research endeavors should recognize and address the significance of optimization for the overall efficacy and sustainability of MSWI operations.

2.2 Could you explain why you chose such structure of the paper?

Response 2.2:

This paper aims to review the application status of artificial intelligence (AI) technology in the MSWI process. In section 2, we introduced the MSWI process and provided the existing states and challenges. In sections 3-6, the existing literature was reviewed respectively according to modeling, control, optimization, and maintenance. In section 7, we discussed the outlook on AI applications for the MSWI process. In section 8, we summarized the entire article. The structure of the paper referred to the study [R1].

[R1] Gallegos, J.; Arévalo, P.; Montaleza, C.; Jurado, F. Sustainable Electrification—Advances and Challenges in Electrical-Distribution Networks: A Review. Sustainability 2024, 16 (2), 698.

2.3 Maybe even a figure could help to reveal the logic of your literature review.

Response 2.3:

In the end of the Introduction section of the revised version, we have provided a figure to show the structure of this study. It is also shown the logic of our literature review, which is shown as follows.

The structure of this study is shown in Figure 2.

Figure 2. Structure of this study

Section 2 introduces the literature review methodology, provides a detailed description of the MSWI process, and offers a brief overview of AI applications for optimal control. Subsequently, sections 3-6 delve into the individual fields of AI application research, addressing modeling, control, optimization, and maintenance of the MSWI process. In Section 7, the focus shifts to an in-depth discussion of the prospects and outlook on AI applications within the MSWI process. Finally, Section 8 encapsulates the key findings and conclusions of this study.

Point 3: there is a complete lack of methodology explanation.

Response 3:

Thanks.

In the Section 2, we introduce the review methodology in detail.

2.1. Methodology about literature review

The literature reviewed in this study was systematically collected and processed from prominent scientific research databases, including WoS, Engineering Village, PubMed, and China National Knowledge Internet (CNKI). The retrieval time range spans from the establishment of each database to December 2023. Subsequently, the collected literature underwent a meticulous filtering process to exclude unrelated works. The refined literature was then categorized into four distinct sections based on AI applications in the MSWI process, namely modeling, control, optimization, and maintenance. Figure 3 illustrates the distribution of the literature across these four categories, providing an overview of the research landscape.

Figure 3. The number of literature in each category

Figure 3 reveals a notable concentration of research in the areas of modeling and control, while comparatively fewer studies focus on the optimization of the MSWI process using AI technology. It is crucial to highlight that optimization plays a pivotal role in achieving the sustainable development of the MSWI process. Despite the current emphasis on modeling and control, future research endeavors should recognize and address the significance of optimization for the overall efficacy and sustainability of MSWI operations

Point 4: there is no quantification of the described statements and no real comparison between the cited studies.

Response 4:

Thanks.

In Sections 3-6, we added tables to clearly show the comparison, as follows:

Table 1. Summary of modeling researches on combustion process

Category

Object

Technology

Benefit

Year

Literature

Key controlled variables modeling

FT

Multi-model intelligent combination

w   Based on the decision tree C4.5, the algorithm can be selected for different data sets to build an ensemble model to improve the accuracy.

2019

[39]

T-S Fuzzy neural network

w   The correlation between FT and input variables is obtained by using the internal weight of the neural network.

2020

[40]

Least squares-support vector regression

w   Based on the principle of minimizing structural risk, the generalization ability and robustness are improved, and the over-fitting problem is avoided.

2023

[41]

FGOC

Long short-term memory network

w   The PSO algorithm is used to optimize the network hyperparameters to improve the model accuracy.

2021

[43]

SF

Radial basis function networks

w   The minimum resource allocation network technology is combined with the adaptive extended Kalman filter to update all the parameters of the network, so as to serve the MPC.

2011

[37]

Radial basis function networks

w   The mean impact value algorithm is used to filter the features, which enhances the robustness and accuracy while reducing the model structure.

2022

[45]

Long short-term memory network

w   The dynamic update of the model based on real-time data improves the prediction accuracy.

2021

[46]

CLP

Neural network

w   Based on the waste quality and quantity in different incinerator types and different seasons, an online learning method is proposed, which can select an optimized neural network.

w   The control accuracy of pollutant emission is improved by using flame combustion image information.

1996

[47]

FGOC and SF

System identification

w   Multiple data sets can be used to improve the model's accuracy to adapt to a variety of working conditions.

2002

[48]

FT, FGOC, and SF

System identification

w   A cascade structure is designed by simulating the actual industrial process.

w   The optimization algorithm is used to identify the parameters and improve the accuracy of the model.

2021

[49]

FT, FGOC, and SF

T-S Fuzzy neural network

w   The complementary information between multiple tasks is used to accurately fit multiple controlled variables at the same time, which improves the dynamic adaptability of the model.

2022

[50]

FT, FGOC, and SF

Decision tree algorithm

w   The integration of RF and GBDT not only simplifies the model dimension but also improves the model accuracy.

2021

[51]

Auxiliary variables modeling

CVMSW

Estimation of waste heat balance

w   The method is simple and easy to calculate.

2017

[56]

Estimation of waste heat balance

w   The variables involved are easy to monitor and the estimated values can be calculated in real time.

2019

[57]

Mass balance

w   It can be directly obtained by online measurement of the gas components CO2, H2O, O2, and the H2O in the surrounding air.

2002

[58]

Back propagation neural network, Radical basis function neural network, and Adaptive neural fuzzy inference system

w   The method is simple, easy to implement, and low-cost.

2016

[59]

Back propagation neural network

w   Based on the correlation analysis, the partial correlation coefficient is obtained, and then the model is established. Compared with the multiple linear regression model, the accuracy is improved.

2002

[60]

Back propagation neural network

w   The accuracy of the model is improved by determining the network hyperparameters upon experiments and analysis.

2003

[61]

Back propagation neural network

w   For inaccurate, contradictory, and erroneous data, the neural network model has stronger fault tolerance than the physical component model, and then obtains higher accuracy results.

2010

[62]

Back propagation neural network

w   The genetic algorithm is used to optimize the network parameters, thereby improving the accuracy.

2012

[63]

L-M backpropagation neural network

w   Based on the element content, the accurate prediction of the high calorific value of waste was realized.

2010

[64]

Fuzzy neural network

w   Based on MI and PSO, the input features of the model are screened to reduce the computational complexity of the model.

2021

[65]

Back propagation neural network, Support vector machine, Adaptive neuro-fuzzy inference system, and Random forest

w   Based on expert experience, the calorific value of waste is classified;

w   The PSO algorithm is used to optimize the model parameters, and then a model with high accuracy is established.

2017

[66]

Least-square support vector machine

w   The genetic algorithm is used to optimize the model parameters.

w   The sensitivity analysis experiment was carried out on the input characteristics. The results show that the percentage of carbon has the deepest influence on HHV prediction.

2018

[67]

Deep learning

w   It is proposed to establish a waste image database to support the combination of image recognition technology and deep learning to achieve calorific value prediction.

2021

[68]

TMSW

Soft sensing model

w   The parameters such as air pressure, negative pressure, grate area, air volume and temperature are used to estimate.

2022

[69]

Soft sensing model

w   The thickness of the MSW layer is estimated based on the MSW composition and grate movement.

2022

[70]

Table 2. Summary of modeling researches on operational indices

Category

Object

Technology

Benefit

Year

Literature

Environmental indices

NOx

System identification

w   It can not only compensate the delay time of the detection device, but also the whole process.

1998

[72]

System identification

w   A continuous-time MISO reduced-order model is constructed.

2002

[73]

Back propagation neural network

w   The number of hidden layer nodes is determined by dynamic construction method.

2004

[75]

Radial basis function neural network

w   The complex task is decomposed into sub-models to obtain a more accurate prediction model.

2020

[76]

Radial basis function neural network

w   The self-organizing and competitive integration strategies are used to construct the sub-model to enhance the generalization performance and efficiency.

2021

[77]

Long short-term memory

w   A cooperative decision strategy is designed to ensure the generalization performance of modular model.

2023

[78]

CO

Long short-term memory

w   The PSO algorithm is used to adaptively reduce depth features and hyperparameters.

2024

[82]

DXN

Numerical modeling

w   The flow-and temperature distribution and the residence-time behavior are obtained.

1989

[89]

Linear regression

w   Dummy variables are included to further provide the selective capability of different process.

1995

[90]

Linear regression

w   Based on the static analysis of the model, suggestions for minimizing DXN concentration are given.

1997

[91]

Back propagation neural network

w   The genetic programming model is used to screen out non-linear models as well as identify the system parameters simultaneously in a highly complex system based on a small set of samples.

2000

[92]

Back propagation neural network

w   The genetic algorithm is used to optimize the parameters to improve the accuracy of the model.

2008

[93]

Support vector regression

w   The inputs of the model are determined based on the mechanism and the correlation analysis of working conditions and conventional pollutants.

2017

[94]

Least squares-support vector machine

w   The optimal selection algorithm based on branch and bound and the information entropy weighting algorithm based on prediction error are used to adaptively select and weigh the candidate sub-models.

2022

[95]

Random forest and gradient boosting decision tree

w   The RF is used to reduce the model dimension, and then the GBDT algorithm is used to improve the model accuracy.

2020

[96]

Random forest

w   The prediction errors are used to cyclically calculate the weight of the source and target domain samples.

2020

[97]

Product index

HRR

Equipment

w   The automatic measurement is realized with less manual intervention, and the analysis efficiency is improved.

2021

[109]

Image recognition

w   Based on the slag image, a reference card of slag color gradient marked with heat reduction rate is generated to guide related operations.

2022

[110]

Table 3. Summary of researches on control in on-site

Category

Object

Technology

Benefit

Year

Literature

ACC system

FT

Thermography-assisted combustion control system

w   It can quickly obtain the temperature distribution in the furnace to reduce the response time, thereby reducing the fluctuation of parameters.

1994

[119]

Whole process

Fuzzy system and Neural network

w   Based on process data and images, a combustion state recognition model is established to assist ACC system decision-making.

1998

[120]

Whole process

Infrared image analysis instrument

w   On-line acquisition and analysis of combustion images are realized.

2006

[121]

Negative pressure

Expert experience

w   While improving the negative pressure monitoring of the furnace to suppress the fluctuation of the negative pressure control system, the automatic control scheme of the leachate recirculation flow control system is designed.

2004

[122]

Whole process

Expert experience

w   The controller performance requirements are low.

2017

[123]

Pollutant

Expert experience

w   The pollution emission data is added to the ACC system to intervene in advance to reduce emissions.

2019

[124]

Non-ACC system

Whole process

Fuzzy logic

w   The fuzzy logic rules of monitoring and control are developed through expert experience.

1989

[125]

FT

Fuzzy logic

w   The adaptability of the incinerator to the calorific value of MSW is improved.

2003

[126]

FT

Fuzzy logic

w   Knowledge is modularized in the form of rules and events to deal with different situations.

2006

[127]

Table 4. Summary of researches about control in off-site

Category

Object

Technology

Benefit

Year

Literature

SISO

FT

Fuzzy logic

w   The real-time estimation of waste water content is realized to compensate the controller action error.

1993

[118]

FT

Fuzzy logic

w   The control rule factor optimized by the correction algorithm improves the ability of the control system to interference responds.

2005

[128]

FT

Fuzzy logic

w   On the basis of summarizing the control rules according to the weight variation law of temperature deviation and temperature deviation change rate, the adaptive weighting factor of input variables is introduced to improve the adaptive ability of the control system.

2004

[129]

FT

Fuzzy logic

w   In order to deal with the problem of rule explosion, a hierarchical fuzzy controller is proposed, and the online learning and correction of control parameters and control rules are realized.

2004

[130]

FT

Fuzzy logic

w   The fuzzy logic is used to automatically adjust the PID parameters, thereby improving the adaptability.

2008

[131]

FT

Radial basic function, and Event-trigger

w   The online adjustment of PID controller parameters is realized based on RBF.

w   The event triggering method is used to reduce the update frequency of the controller.

2022

[132]

FT

Human-simulated intelligent controller

w   The control strategy is closer to the actual needs of the project.

2013

[133]

FT

Human-simulated intelligent controller

w   The overshoot can be effectively suppressed for uncertain disturbances.

2015

[134]

FT

Human-simulated intelligent controller

w   No need for accurate theoretical model support

2016

[135]

FT

Human-simulated intelligent controller

w   PSO is used to tune the controller parameters.

2018

[136]

FGOC

Radial basis function, Model predictive control

w   The adaptive fuzzy C-means is used to determine the network parameters.

w   The prediction model parameters are adjusted online by an adaptive update strategy.

2023

[137]

SF

Fuzzy logic

w   It has the potential of about a 10% increase in the capacity of MSW processing and electricity generation.

1995

[138]

SF

Fuzzy logic

w   The controller can be easily and simply deployed.

w   It can effectively suppress the uncertainty disturbance.

2000

[139]

SF

PI Controller

w   A periodic control strategy is proposed for the feed characteristics of the incinerator.

2003

[140]

SF

Linear quadratic regulator

w   A full-state closed-loop feedback loop with integral loop is designed.

2020

[141]

MIMO

SF, and FGOC

Linear model predictive control

w   It can effectively suppress the influence of large disturbance on the control system.

2005

[142]

SF, and FGOC

Nonlinear model predictive control

w   A moving horizon estimator is used to estimate the states and disturbances.

2005

[143]

SF, and FGOC

PID controller

w   Add disturbance rejection loops to improve controller performance.

2010

[144]

FT, and FGOC

Fuzzy neural network

w   The controller structure is self-organized and adjusted by calculating the similarity of neurons and multi-task learning ability.

2023

[145]

FT, SF, and FGOC

PID controller

w   A quasi-diagonal recurrent neural network is used to adjust the control parameters automatically.

2022

[146]

FT, SF, and FGOC

Single neuron adaptive PID controller

w   A multivariable serial control structure is designed based on the process flow.

2023

[147]

Table 5. Summary of optimization researches

Object

Technology

Benefit

Year

Literature

AD

Case-based reasoning

w   The effectiveness of the proposed method is verified in the simulation platform.

2020

[148]

Case-based reasoning, random weight neuron network, and radial basis function

w   The optimal setting of secondary air volume is realized by integrating multiple intelligent algorithms.

2022

[149]

Multi-objective particle swarm optimization

w   Based on the population state, the corresponding update methods are designed to improve the problem of falling into local optimum.

2023

[150]

Multi-objective particle swarm optimization

w   An adaptive knowledge transfer strategy is designed to improve optimization efficiency.

2023

[151]

MD

Multi-objective genetic algorithm

w   FLIC is used to generate model training data.

w   It has strong expansibility and portability.

2005

[152]

CV

Multi-objective competitive swarm optimization

w   A comprehensive nondominated evaluation system and improved competitive mechanism are proposed.

w   An adaptive scheme combined with multi-strategy learning is proposed.

2024

[153]

Table 6. Summary of maintenance researches

Object

Technology

Benefit

Year

Literature

Recognition of flame status

Multi-scale color moment features and random forest

w   Certain interpretability.

w   Combining local and global features.

2019

[156]

Generative adversarial network

w   Increasing the number of modeling samples.

2022

[157]

DFC based on convolutional multi-layer feature fusion

w   Developing an online combustion status-recognition platform.

w   Integrating deep fusion features with the DFC.

2023

[158]

Qualitative detection of operational fault

Fuzzy expert system

w   Good visualization.

w   Strong practicality.

1994

[165]

Cluster analysis, artificial neural networks, and Monte Carlo simulation

w   Accurate state monitoring for steam generation and NOx control.

2008

[166]

Fault tree and expert system

w   Continued updates.

2008

[167]

Back propagation neural network

w   Integrating multiple neural networks.

2008

[168]

Back propagation neural network

w   High accuracy.

2015

[169]

Radom weight neuro network and case-based reasoning

w   Reduction of the detection time complexity.

2021

[170]

Quantitative detection of operational fault

Principal component analysis

w   Earlier than human operators.

w   Reducing the misreporting rate.

2008

[174]

Principal component analysis and partial least square

w   Good performance in fault detection and isolation.

2011

[175]

Point 5: many concepts are stated with few references’ multiple times, even if the list of references is appropriate.

Response 4:

Thanks.

In the revised revision, we added some references to explain the related concepts, and explained the MSWI process related nouns. For example,

Figure 4 depicts the process flow of the grate-type MSWI process in Beijing.

Figure 4. Process flow of an MSWI plant in Beijing

Note: Flue Gas1 denotes the flue gas at the furnace outlet. Flue Gas2 represents the flue gas at the inlet position of the induced draft fan. Flue Gas3 corresponds to the flue gas at the chimney outlet. Within the existing body of research, the predominant focus has been on Flue Gas1 and Flue Gas3.

Key controlled variables in the combustion process encompass furnace temperature (FT), flue gas oxygen content (FGOC), steam flow (SF), and combustion line position (CLP), which refers to the position where the end of MSW becomes ash [34], et al.

Moreover, we have added a table of abbreviations and their corresponding meanings at the end of the article. It is shown as follows.

Table S1 Abbreviations and their meanings

Abbreviations

Meanings

AI

Artificial intelligence

IoT

Internet of things

MSW

Municipal solid waste

MSWI

Municipal solid waste incineration

WTE

Waste-to-energy

WoS

Web of Science

CNKI

China National Knowledge Internet

SNCR

Selective non-catalytic reduction

PSO

Particle swarm optimization

PCA

Principal component analysis

PLS

Partial least squares

NN

Neural network

RBFNN

Radial basis function neural network

MNN

Modular neural network

LS-SVM

Least-square support vector machine

DBN

Deep belief network

Yolo

You only look once

DFR-clfc

Deep forest regression based on cross-layer full connection

IDFR

Improved deep forest regression

SVM

Support vector machine

TM

Tree-based model

FL

Fuzzy logic

FNN

Fuzzy neural network

DL

Deep learning

FT

Furnace temperature

FGOC

Flue gas oxygen content

SF

Steam flow

CLP

Combustion line position

MISO

Multi-input single-output

LS-SVR

Least squares-support vector regression

LSTM

Long short-term memory network

RBF

Radial basis function

MIMO

Multi-input multi-output

RF

Random forest

GBDT

Gradient boost decision tree

CVMSW

Calorific value of municipal solid waste

ANFIS

Adaptive network based fuzzy inference system

ANN

Artificial neural network

TMSW

Thickness of the municipal solid waste layer

CEMS

Continuous emission monitoring system

DXN

Dioxin

VOCs

Volatile organic compounds

CO

Carbon monoxide

BPNN

Back propagation neural network

SVR

Support vector regression

APCDs

Air pollution control devices

HRR

Heat reduction rate of slag

ACC

Automatic combustion control

SISO

Single-input and single-output

HSIC

Human simulated intelligent controller

LMPC

Linear model predictive control

NMPC

Nonlinear model predictive control

PID

Proportional integral differential

DCS

Distributed control system

AD

Air distribution

MD

Material distribution

CBR

Case-based reasoning

CV

Controlled variables

GANs

Generative adversarial networks

DFC

Deep forest classification

CBR

Case-based reasoning

RWNN

Random weight neural network

MSPM

Multivariate statistical process monitoring

VSG

Virtual sample generation

Point 6: Figures from previous studies should be cited.

Response 6:

Thanks.

After revising the manuscript, we added a reference to the relevant graph.

For example:

Figure 9. The detection process of DXN concentration [88]

[88] Xia, H., Tang, J., Yu, W. Qiao, J. F. Online Measurement of Dioxin Emission in Solid Waste Incineration Using Fuzzy Broad Learning. IEEE Transactions on Industrial Informatics 2024, 20 (1), 358–368.

Point 7: I understand the effort before a paper submission, and I regret to write that I do not see enough novelty and deep analysis compared to the long reference list. I would focus more on the review method and the literature comparison.

Response 7:

Thanks.

The purpose of this study is to review the existing research on the MSWI process, to find the defects of the existing research and the future development prospects.

According to your suggestion, we rearranged and restructured the manuscript.

In the new version, we have added the description of the review methods and the literature comparison tables. At the same time, the timeline diagram of the application of AI technology in the MSWI process is given to clearly show the development and changes at the first time.

Round 2

Reviewer 5 Report

Comments and Suggestions for Authors

The revised version of the paper looks better now. The paper could be accepted for publication in its current form.